# Towards a molecular mechanism underlying mitochondrial protein import through the TOM and TIM23 complexes

Holly C Ford, William J Allen, Gonçalo C Pereira[†], Xia Liu, Mark Simon Dillingham, Ian Collinson*

School of Biochemistry, University of Bristol, Bristol, United Kingdom

**Abstract** Nearly all mitochondrial proteins need to be targeted for import from the cytosol. For the majority, the first port of call is the translocase of the outer membrane (TOM complex), followed by a procession of alternative molecular machines, conducting transport to their final destination. The pre-sequence translocase of the inner membrane (TIM23-complex) imports proteins with cleavable pre-sequences. Progress in understanding these transport mechanisms has been hampered by the poor sensitivity and time resolution of import assays. However, with the development of an assay based on split NanoLuc luciferase, we can now explore this process in greater detail. Here, we apply this new methodology to understand how $\Delta\psi$ and ATP hydrolysis, the two main driving forces for import into the matrix, contribute to the transport of pre-sequence-containing precursors (PCPs) with varying properties. Notably, we found that two major rate-limiting steps define PCP import time: passage of PCP across the outer membrane and initiation of inner membrane transport by the pre-sequence – the rates of which are influenced by PCP size and net charge. The apparent distinction between transport through the two membranes (passage through TOM is substantially complete before PCP-TIM engagement) is in contrast with the current view that import occurs through TOM and TIM in a single continuous step. Our results also indicate that PCPs spend very little time in the TIM23 channel – presumably rapid success or failure of import is critical for maintenance of mitochondrial fitness.

## Editor's evaluation

In this study, a bioluminescence-based technique is used to analyze the import of precursor proteins into the mitochondrial matrix in real time. This is an innovative technical advance that can provide mechanistic details of the kinetic steps involved in mitochondrial protein import. It may potentially be used for other membrane protein transport systems and for drug screening studies targeting the mitochondrial import apparatus.

## Introduction

Mitochondria are double membrane-bound eukaryotic organelles responsible for the biosynthesis of ATP among many other essential cellular functions (*Nowinski et al., 2018*; *Rouault, 2012*; *Nicholls, 1978*; *Chen et al., 2003*; *Nishikawa et al., 2000*; *Hoth et al., 1997*; *Chandel, 2015*; *Wang and Youle, 2009*). Of more than a thousand proteins that constitute the mitochondrial proteome, all but a handful (encoded on the mitochondrial genome – 13 in human) are synthesised in the cytosol and must be imported. Almost all mitochondrial proteins (exceptions include precursors of α-helical outer mitochondrial membrane [OMM] proteins) initially enter mitochondria via the translocase of the outer membrane (TOM complex) which contains the pore-forming β-barrel protein Tom40 (*Ahting et al.,*

*For correspondence:
ian.collinson@bristol.ac.uk

Present address: †Mitochondrial Biology Unit, University of Cambridge, Cambridge, United Kingdom

Competing interest: The authors declare that no competing interests exist.

*2001*; *Guan et al., 2021*; *Araiso et al., 2019*). From here, they are delivered to a number of bespoke protein import machineries, which direct them to their final sub-mitochondrial destination: the OMM, inter-membrane space (IMS), inner membrane (IMM), or the matrix.

Roughly, 60–70% of mitochondrial precursor proteins – almost all those targeted to the matrix and a subset of IMM proteins – have a positively charged, amphipathic α-helical pre-sequence, also known as a mitochondrial targeting sequence (MTS; *Araiso et al., 2019*; *Vögtle et al., 2009*). Following emergence from the Tom40 channel, these pre-sequence-containing precursors (PCPs) are transferred to the translocase of the inner membrane (TIM23-complex), through which they pass in an unfolded state (*Eilers and Schatz, 1986*; *Matouschek et al., 1997*; *Neupert and Brunner, 2002*; *Rassow et al., 1990*; *Neupert and Herrmann, 2007*). Genetic and biochemical experiments have elucidated the key constituents of the TIM23-complex (*Blom et al., 1993*; *Maarse et al., 1992*; *Emtage and Jensen, 1993*; *Maarse et al., 1994*). The core (TIM23$^{CORE}$) comprises three membrane-spanning proteins: Tim23, Tim17, and Tim50 and associates with different proteins to form complexes tailored for different tasks. Together with Tim21 and Mgr2, it forms the TIM23$^{SORT}$ complex, capable of lateral release of proteins with hydrophobic sorting sequences. Association with the pre-sequence translocase-associated motor (PAM) forms the TIM23$^{MOTOR}$ complex, responsible for matrix import.

Our current understanding of protein import via the TOM and TIM23$^{MOTOR}$ complexes is summarised in *Figure 1A*. After entry of the PCP through TOM, the electrical component of the proton-motive force (PMF) across the IMM – the membrane potential ($\Delta\psi$; negative in the matrix) – is required, acting as an electrophoretic force on the positively charged pre-sequence (*Martin et al., 1991*; *Geissler et al., 2000*; *Truscott et al., 2001*). $\Delta\psi$ alone is sufficient for insertion of membrane proteins via the TIM23$^{SORT}$ complex (*Callegari et al., 2020*), but complete import into the matrix by the TIM23$^{MOTOR}$ complex requires an additional driving force: ATP hydrolysis by the main component of PAM, the mtHsp70 protein (Ssc1 in yeast) (*Wachter et al., 1994*), which pulls the rest of the PCP through to the matrix after the MTS has been imported. Finally, following passage through or into the IMM, the MTS is cleaved by a matrix processing peptidase (*Vögtle et al., 2009*).

The above model is primarily derived from end point measurements of classical import assays reported by autoradiography or Western blotting. However, this method is limited in its time resolution, and insufficient to provide a deep understanding of the individual steps that make up import, or their relative contributions to its kinetics. For this reason, we recently developed a highly time-resolved and sensitive assay which exploits a split NanoLuc enzyme (*Pereira et al., 2019*; *Dixon et al., 2016*) to measure protein transport across membranes (*Figure 1B*). In the NanoLuc assay, PCPs tagged with a small fragment of the NanoLuc enzyme (an 11 amino acid peptide called pep86, engineered for high affinity) are added to mitochondria isolated from yeast engineered to contain a matrix-localised large fragment of the enzyme (the enzyme lacking a single β-strand, called 11S). When the PCP-pep86 fusion protein reaches the matrix, pep86 binds rapidly and with tight affinity to 11S forming a complete NanoLuc luciferase. In the presence of the NanoLuc substrate (Nano-Glo luciferase assay substrate), this generates a luminescence signal proportional to the amount of NanoLuc formed. Luminescence is thus a direct readout of the amount of pep86 (and hence PCP) that has entered the matrix, up to the total amount of 11S. As expected, it is $\Delta\psi$-dependent, affected by depletion of ATP, and sensitive to specific inhibitors of TIM23-dependent protein import (*Pereira et al., 2019*).

Here, we continue the use of the NanoLuc translocation assay to obtain precise, time-resolved measurements of protein delivery into the matrix mediated by the TOM and TIM23$^{MOTOR}$ complexes. Its application to accurately measure both rates and end point values (amplitude) turn out to be critical for the development of a model for import. Thus, to add mechanistic detail to the above model (*Figure 1A*), we systematically varied the length and charge of the mature sequences of PCPs and profiled their import kinetics. To better understand the cause of any effects on the observable kinetic parameters (amplitude, rate, and lag), we performed experiments under conditions where either of the two main driving forces, $\Delta\psi$ or ATP, had been depleted.

Our results suggest that IMM transport itself is fast in normally functioning mitochondria and limited by the availability of $\Delta\psi$. The rate of import is instead limited by transport across the OMM, which is strongly dependent on protein size, and initiation of transport across the IMM by the MTS. Analyses such as these, together with emerging structures of the import machinery (e.g., *Tucker and Park, 2019*), will be fundamental to our understanding of the underlying molecular basis of mitochondrial protein import.

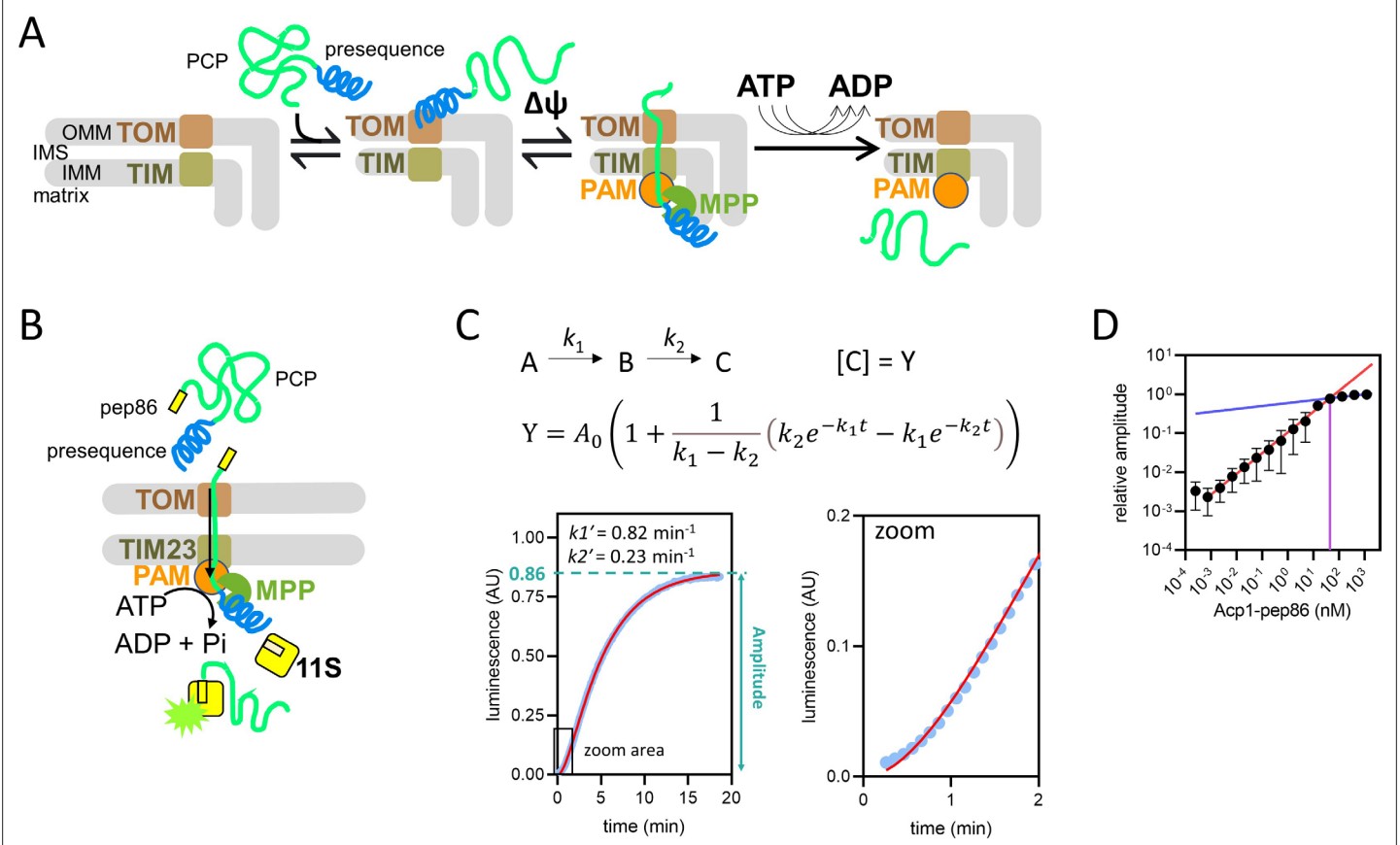

**Figure 1.** Model of pre-sequence-containing precursor (PCP) import into mitochondria and outline of the NanoLuc import assay. (**A**) Simple model of PCP import into mitochondria, showing binding of PCP to the translocase of the outer membrane (TOM complex), $\Delta\phi$-dependent movement of the pre-sequence into the matrix, and ATP-dependent translocation of the remainder of the protein. (**B**) Diagramatic representation of the NanoLuc real-time import assay, which is essentially the model in (**A**) plus the binding of the C-terminal pep86 to internalised 11S, which forms NanoLuc in the matrix. (**C**) An example of luminescence data from the NanoLuc import assay of 1 µM DDL (one of the length variant PCPs, see Results) in energised mitochondria, showing the fit to a model for two consecutive, irreversible steps (see Methods). The final step gives rise to signal such that [C] (concentration of C) is proportional to luminescence. The order of the two steps is assigned arbitrarily. (**D**) The effect of varying PCP concentration (Acp1-pep86) on amplitude of signal from import reactions. A straight line was fitted to the data where amplitude increased linearly with PCP concentration (red) and to the data where amplitude increased only marginally (blue). The intersect of these lines and corresponding PCP concentration (~45 nM), the point of plateau, is also shown (purple). Data are the mean ± SD of three independent biological experiments.

The online version of this article includes the following source data and figure supplement(s) for figure 1:

**Source data 1.** Numerical data corresponding to the example luminescence data and fit in panel C.

**Source data 2.** Numerical primary (luminescence) and secondary (amplitudes) data corresponding to the graph in panel D.

**Figure supplement 1.** 11S levels and signal amplitude.

**Figure supplement 1—source data 1.** Unprocessed image of the Western blot in panel A.

**Figure supplement 1—source data 2.** Numerical data corresponding to the luminescence traces in panel B.

**Figure supplement 1—source data 3.** Densitometry values for the bands in the Western blot in panel A, calculations of matrix 11S concentration, and numerical data corresponding to the graph in panel C.

**Figure supplement 1—source data 4.** Data corresponding to panel D.

**Figure supplement 2.** Constraints of data fitting to the NanoLuc import traces.

**Figure supplement 2—source data 1.** Numerical primary (luminescence) and secondary (rate and amplitude values) corresponding to the graph in panel B.

**Figure supplement 2—source data 2.** Numerical data corresponding to the (normalised) luminescence traces in panel C.

## Results

### The import reaction is largely single turnover under the experimental conditions deployed

Initially we set out to verify that bioluminescence is a true measure of matrix import. An exemplar NanoLuc import trace is shown in *Figure 1C*, collected using the precursor of the model yeast matrix protein Acp1 (also used in previous import studies *Wurm and Jakobs, 2006*) fused to pep86 (Acp1-pep86). The most striking parameter of this trace is amplitude (see below for full fitting details), which corresponds to the amount of NanoLuc formed when the reaction reaches completion (the end point), and thus the total number of import events; provided the pep86 tag does not exceed matrix 11S. In order to verify that this was not the case, we estimated the concentration of 11S in the mitochondria by quantitative Western blotting. An antibody raised against intact NanoLuc was used to compare the quantities of mitochondrial 11S with known amounts of purified protein (*Figure 1—figure supplement 1A*). The results revealed high (µM) internal 11S concentrations with some variation between mitochondrial preparations (~2.8–7.5 µM; see source data). Analysis of the import of saturating PCP into the different mitochondrial preparations demonstrated that the amounts of 11S and signal amplitude are not correlated, i.e., lower internal concentrations of 11S do not elicit lower amplitudes (*Figure 1—figure supplement 1B, C*). This confirms that signal amplitude is not limited by 11S (see also below) and is a faithful measure of imported PCP, irrespective of how much is added to the outside.

Matrix-localised 11S migrates somewhat slower (higher) on a gel compared to purified 11S (*Figure 1—figure supplement 1A*). While conceivable that this divergence in migration between the two forms results from differing SDS-binding (common for β-barrel proteins such as 11S), it is also possible that the MTS of the matrix-localised version has not been removed. To confirm its matrix localisation, we measured the extraction of known IMS and matrix proteins and 11S itself in response to treatment of mitochondria with increasing concentrations of digitonin. As expected, the IMS protein Tim10 is very easily released by digitonin treatment, while the matrix marker glutathione reductase and 11S are more resistant (*Figure 1—figure supplement 1D*). Thus, we can conclude that 11S is correctly localised in the matrix and not the IMS.

We next measured signal amplitude over a wide range of concentrations of Acp1-pep86. Plotting the results shows that amplitude is linearly related to PCP concentration from 753 fM up to ~45 nM, where it plateaus (*Figure 1D*). Because the mitochondrial matrix volume is only ~1/12,000 of the total reaction volume (see Methods), if all 45 nM PCP were imported, it would correspond to roughly 540 µM inside the matrix. This would be far in excess of the internal 11S concentration (as low as ~2.8 µM), which we know not to be the case (see above). It is also implausible simply from the amount of physical space available. Evidently then, only a tiny fraction of the PCP added reaches the matrix.

As neither the amount of PCP added nor the amount of 11S in the matrix appears to be limiting, we next tested to see whether the number of import sites might be having an effect. To estimate the number of import sites, we generated a PCP that can enter and give a signal, but which prevents subsequent import events through the same site – forcing single turnover conditions. To do this, we fused dihydrofolate reductase (DHFR) to a model PCP; in the presence of the inhibitor methotrexate (MTX), DHFR folds tightly and cannot be imported (*Pfanner et al., 1987*; *Gold et al., 2017*). As expected, if DHFR is omitted (PCP-pep86), MTX has no effect (*Figure 2A*, grey bars); while if it is positioned N-terminal to pep86 (PCP-DHFR-pep86), we see very little luminescence with MTX present – indicative of an efficient import block (*Figure 2A*, blue bars). The efficacy of the MTX block is also confirmed by classical Western blotting import assays (*Figure 2—figure supplement 1* – import failure indicated by red box). However, when DHFR is positioned C-terminal to pep86 (PCP-pep86-DHFR) with sufficient length between the two to span the TOM and TIM complexes (212 amino acids in this case, longer than the 135 required [*Rassow et al., 1989*]), we do see an import signal (*Figure 2A*, orange bars). This confirms that NanoLuc can form as soon as pep86 enters the matrix and does not require the entire PCP to be imported, as seen previously with the bacterial Sec system (*Allen et al., 2020*).

Importantly, the presence or absence of MTX makes only a minor difference to the amplitude of this signal (*Figure 2A*, orange bars). Indeed, the signal amplitude as a function of the [PCP-pep86-DHFR] is similar in the presence or absence of MTX (*Figure 2B*). The slope, which corresponds to the increase in amplitude per 1 nM PCP-pep86-DHFR, is 1.22 times greater in the absence of MTX,

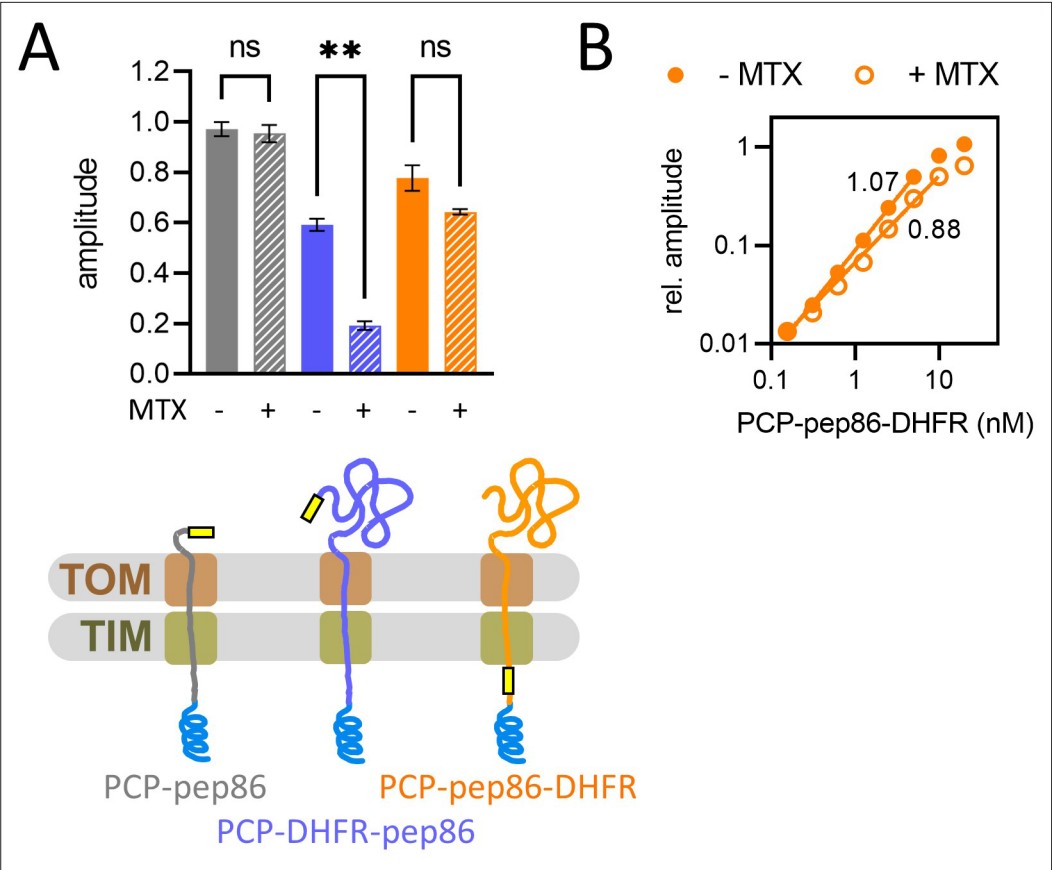

**Figure 2.** Basic characterisation of pre-sequence-containing precursor (PCP) import and turnover number. (**A**) The effect of methotrexate (MTX) on signal amplitude of three proteins (depicted schematically below): PCP-pep86 (grey), for which MTX should have no effect; PCP-dihydrofolate reductase (DHFR)-pep86 (blue), where MTX prevents entry of pep86; and PCP-pep86-DHFR (orange), where MTX limits import to one pep86 per import site. Bars show the average and SEM from three independent biological replicates. Differences between groups were analysed using a one-way ANOVA test, with Geisser-Greenhouse correction applied, followed by the Holm-Sidak multiple comparisons test. **, p value 0.0038; ns, not significant. (**B**) Signal amplitude as a function of PCP-pep86-DHFR concentration in the absence (solid circles) and presence (open circles) of MTX.

The online version of this article includes the following source data and figure supplement(s) for figure 2:

**Source data 1.** Numerical primary (luminescence) and secondary (amplitude) data corresponding to the bar chart in panel A.

**Source data 2.** Primary and secondary numerical data corresponding to the graph in panel B.

**Figure supplement 1.** Confirmation, by Western blotting, of efficient blocking of pre-sequence-containing precursor (PCP)-pep86-dihydrofolate reductase (DHFR) import by methotrexate (MTX) and estimation of amount imported in the absence of MTX.

**Figure supplement 1—source data 1.** Western blot acquisition files.

meaning only about 20% of the signal arises from turnovers beyond the first one. Of course, while this does not mean that import is strictly single turnover – which would certainly seem implausible for fully functional mitochondria in their native environment – it does suggest that it behaves as single turnover under the restrictive conditions here, using isolated mitochondria (without the cytosol). Consistent with this interpretation, the amount of PCP-pep86-DHFR that was fully internalised in the matrix in the absence of MTX (11 pmol per mg mitochondria, based on quantification of Western blot bands in *Figure 2—figure supplement 1*) matched closely the approximate amount of TIM23 dimer estimated to be in the sample of mitochondria (8.5 pmol per mg mitochondrial protein [*Sirrenberg et al., 1997*]). It has previously been shown that signal amplitude can be reduced by depleting $\Delta\psi$ (*Pereira et al., 2019*), which would suggest that available energy limits protein import. This

can be reconciled with the apparent single turnover nature of the reaction if 'resetting' the channel after import – possibly through dimerisation of TIM23, as previously reported (*Bauer et al., 1996*) – requires additional energy input.

## Kinetic analysis of import suggests two major rate-limiting steps

In addition to the amplitude values, the import traces contain information about the kinetics of the reaction. Looking again at the data in *Figure 1C*, it can be seen that import does not start at its maximum rate; rather there is a lag before import accelerates. This is characteristic of reactions with multiple consecutive steps, where only the last one gives rise to a signal. As an approximation, the data fitted well to an equation for a two-step process where the second gives rise to the signal (*Figure 1C*; A→B→C, see also Methods), which gives two apparent rate constants ($k_1'$ and $k_2'$) in addition to amplitude (*Fersht, 1984*). Close inspection of the data (*Figure 1C*, right panel) suggests that adding additional steps would marginally improve the fit; however, these additional rate constants would be fast and poorly defined; two steps therefore represent a reasonable compromise between accuracy and complexity.

In the simplest case possible, where the two steps are irreversible and have very different values, $k_1'$ and $k_2'$ correspond to the two rates for these steps ($k_1$ and $k_2$; *Fersht, 1984*). This is complicated if the reactions are reversible (in which case the reverse rates also factor) or if $k_1$ and $k_2$ are very similar (where they are both convoluted into $k_1'$ and $k_2'$). In spite of these potential complexities, the analysis is very useful for understanding the mechanism of import (see below).

It should be noted that, because we have no information for the concentration of the intermediates, the order of the two steps cannot be determined *a priori*. However, as detailed below, they can be distinguished by perturbing the system and seeing how this affects the different rates. From this, and based on the results in the following sections, we assign $k_1'$ as transport of the PCP through TOM and $k_2'$ as subsequent engagement of the MTS with the TIM23 complex.

It is also important to note that any additional step faster than about 5 min$^{-1}$ will not be resolved in our experimental set up – due to the limitations of the plate reader (see detailed explanation in *Figure 1—figure supplement 2A*). Instead, the extra step will manifest itself as a small apparent additional lag before the signal appears (equal to $1/k_{step}$, where $k_{step}$ is the rate constant for that process; *Allen et al., 2020*). This includes formation of NanoLuc: it is >7.4 min$^{-1}$ even at the lowest estimated 11S concentration, as determined in solution (*Figure 1—figure supplement 2B*). Consistent with this, we observe that the import kinetics are not appreciably affected by the concentration of matrix 11S, despite its variance (*Figure 1—figure supplement 2C*).

## Import is dependent on total protein length

To begin to validate what physical processes the two apparent rates correspond to, we first designed and purified two series of four PCPs, varying either in total length or in the N- to C-terminal positioning of pep86 (*Figure 3A*). The length variants all similarly contained the pre-sequence of Acp1 followed by the Acp1 mature domain, with pep86 (L) at the C-terminus. Increase in length was achieved by repeating the mature part of Acp1 up to three times. In between each Acp1 mature domain, we included a scrambled pep86 sequence (D), which does not interact with 11S (*Allen et al., 2020*), such that each tandem repeat has the same overall amino acid (aa) composition.

The length variant set was designed to reveal PCP length-dependence of any import step. Members of the other set (position variants) were all identical to the longest length-variant PCP (four tandem repeats), but with the active pep86 (L) in different positions. Because the position variants (abbreviated as LDDD, DLDD, DDLD, and DDDL) are identical save for the number of amino acids that must enter the matrix before the NanoLuc signal arises, all transport steps (including passage through TOM) should be the same for the whole set. Any differences in their import kinetics must therefore arise from the time it takes them to pass through TIM23 and not the steps prior to that. Note that as shown above (*Figure 2A*) and previously (*Allen et al., 2020*), localisation to an internal loop does not compromise the ability of pep86 to interact with 11S.

Import of all four length variants (L, DL, DDL, and DDDL) and position variants (LDDD, DLDD, DDLD, and DDDL) was monitored at high concentration (1 µM) – saturating for all parameters, see below (*Figure 3—figure supplement 1*). In all cases, the data fit well to the simple two-step model, giving an amplitude and two apparent rate constants, with the faster one assigned as $k_1'$ and the

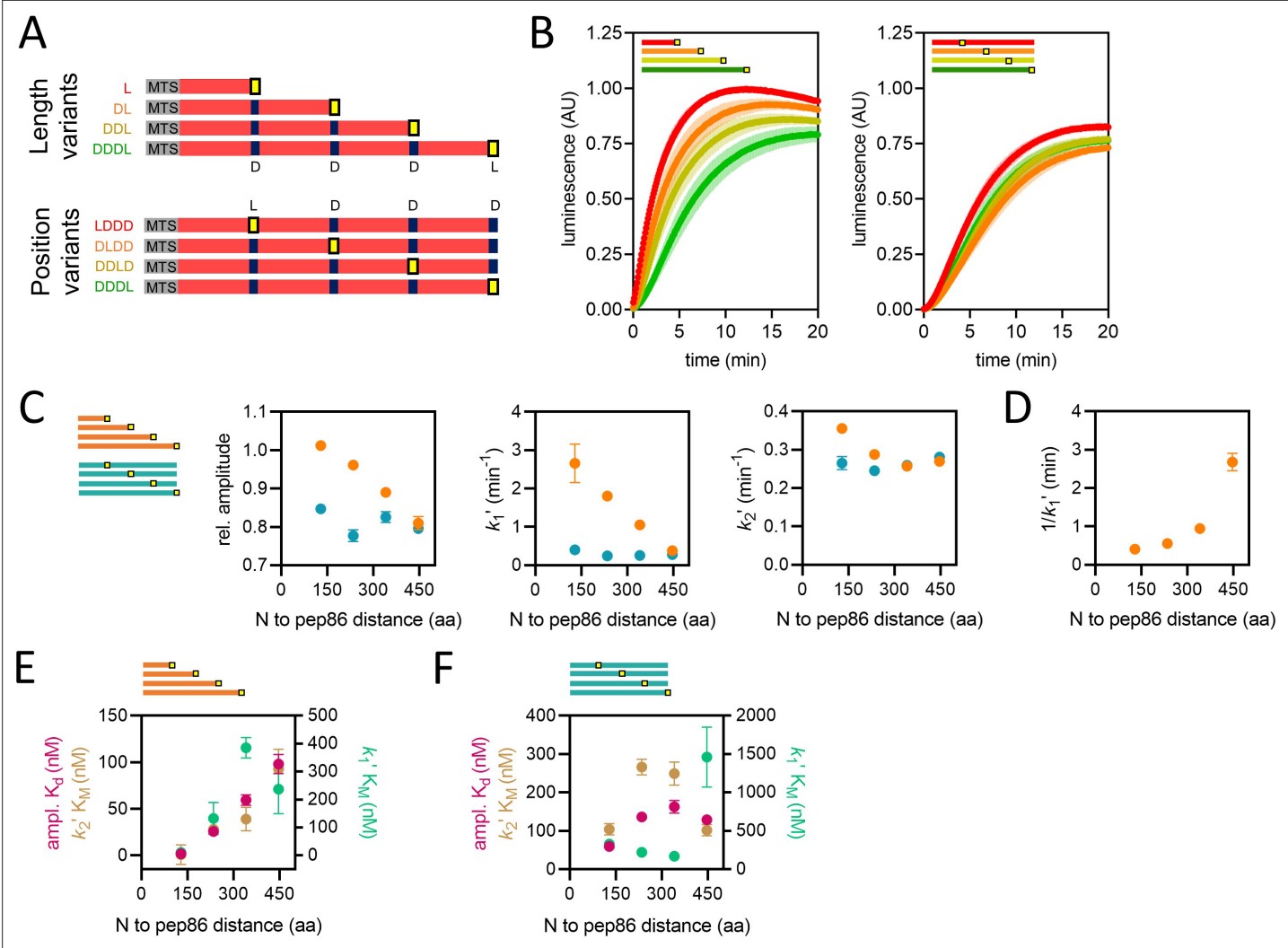

**Figure 3.** Using proteins of varying lengths to elucidate import kinetics. (**A**) Schematic of two protein series (length variants and position variants), with native mitochondrial targeting sequence (MTS) and mature part of Acp1 in grey and red, respectively, pep86 in yellow (L for live) and scrambled pep86 in dark blue (D for dead, i.e., it does not complement 11S). (**B**) Examples of import traces for length variants (left panel) and position variants (right panel). Error bars shown partially transparent in the same colours as the main traces. Those smaller than the main trace are not shown. SD from biological triplicate, each conducted in duplicate. (**C**) Parameters obtained from two-step fits to the data shown in panel **B**. The length variant series is shown in orange and the position variant series in teal. Error bars show SEM from three independent biological experiments, each conducted in duplicate. Error bars smaller than symbols are not shown. (**D**) Reciprocal of $k_1'$ as a function of PCP length (same data as in panel **C**) – the time constant for that step – for the length variants. (**E**) The concentration dependence of length variants. Secondary data from import assays with varying concentrations of length series proteins (four to six independent biological replicates) were fitted to the Michaelis-Menten equation, from which apparent $K_d$s and $K_M$s are derived. Error represents the SEM of this fitting. (**F**) As in panel **E** but with the position variant proteins.

The online version of this article includes the following source data and figure supplement(s) for figure 3:

**Source data 1.** Numerical data corresponding to the luminescence traces in panel B.

**Source data 2.** Numerical data corresponding to the graphs in panel C (secondary data from *Figure 3—source data 1*).

**Source data 3.** Numerical data corresponding to the graph in panel D.

**Source data 4.** Numerical data corresponding to the graphs in panels E and F.

**Figure supplement 1.** The concentration dependence of length and position variants.

**Figure supplement 1—source data 1.** Numerical primary (luminescence) data corresponding to the graphs in panels A–C.

**Figure supplement 1—source data 2.** Numerical primary (luminescence) data corresponding to the graphs in panels D–F.

**Figure supplement 1—source data 3.** Numerical secondary data corresponding to the graphs in panels A–F.

other as $k_2'$. Import traces and the results of fits to the two-step model are plotted in *Figure 3B and C*, respectively. We observe no significant difference between any of the four position variants with respect to any of the three parameters, indicating that transport through TIM23 is fast, and not rate-limiting; therefore, it does not contribute appreciably to the kinetics of import.

For the length series, signal amplitude is inversely correlated with protein length (*Figure 3C*, left panel in orange). Let us suppose that, at any point during processive translocation, an import site can become compromised; for instance, by a PCP becoming trapped in the channel. In this scenario, it would be reasonable to expect a longer protein to have a higher chance of failing to reach the matrix. However, if this was the cause of the dependence of signal amplitude on protein length, we would expect a similar dependence for the position variants, which is not the case (*Figure 3C*, in teal). Nor is this an effect of differing affinities of the PCPs for the initial binding site, as these measurements were performed at saturating PCP concentration. The only plausible explanation is that shorter proteins are able to accumulate at higher levels in the matrix compared to large ones. This observation, that the amplitude varies between different constructs, confirms the important conclusion (made above) that the amount of 11S is in excess and does not limit the import signal; if this was the case then the maximum amplitude would be invariant for all proteins.

Strikingly, we find that $k_1'$ has a strong inverse correlation with PCP length (but not pep86 position), i.e., it is faster for smaller proteins (*Figure 3C*, middle graph). The likely explanation for this is that $k_1'$ corresponds to transport of the entire length of the protein across a membrane. Even more surprisingly, the corresponding step time ($1/k_1'$) increases not linearly but exponentially as a function of PCP length (*Figure 3D*). This means that longer PCPs complete step $k_1'$ more slowly per amino acid. Exponential length-dependence is not a characteristic of a powered or biased directional transport, such as we have seen previously for the Sec system (*Allen et al., 2020*), but rather an unbiased reversible diffusion-based (passive) mechanism (*Simon et al., 1992*). For $k_2'$, meanwhile, there is little difference between the variants (*Figure 3C*, right panel); indeed, with the exception of L, good fits can be obtained when $k_2'$ is fixed globally. Unlike $k_1'$, therefore, $k_2'$ probably corresponds to something other than transport across a membrane.

## Concentration dependence of the two major rate-limiting steps of import

A simple way to assign rate constants to specific events is to measure their dependence on PCP concentration: only steps that involve association between PCP upon the initial contact with the import machinery (with the TOM complex) should show any concentration effect. Thus, we measured protein import for both the length and position variants over a range of PCP concentrations ([PCP]) and fitted the data to the two-step model. Next, we plotted the concentration dependence of each of the three resulting parameters and fitted them to a weak binding (amplitude) or Michaelis-Menten ($k_1'$ and $k_2'$) equation (*Figure 3E–F*; *Figure 3—figure supplement 1*). It should be noted that the $K_M$ values are rough estimates only, as $k_1'$ and $k_2'$ are difficult to precisely assign.

Unexpectedly, all three parameters show a dependence on [PCP] for the length series. The apparent $K_M$s for $k_1'$, (*Figure 3E*, teal) are in the low 100 s of nM and not systematically dependent on PCP length – this is reasonable for initial association of PCP with TOM. The $K_d$s for amplitude and $K_M$s for $k_2'$, meanwhile (magenta and brown, respectively in *Figure 3E*), are very similar to one another: they are very low (high affinity) but increase (decrease in affinity) with increasing PCP length. Because amplitude and $k_2'$ behave identically, it seems reasonable to assume that they reflect the same process, i.e., the final kinetic step of transport (because amplitude is, by definition, successful transport). The precursor length-dependence means that, effectively, longer PCPs require a higher concentration to reach maximum amplitude (*Figure 3E*), even though that amplitude is lower (*Figure 3B–C*). It is not entirely clear why this might be the case: it could reflect a lower affinity of the longer PCPs for TIM23, i.e., the MTS is more effective for shorter proteins. Alternatively, it could be due to a higher propensity of the longer PCPs to form import-incompetent conformations (in the absence of cytoplasmic chaperones) or some other unknown aspect of the energetics of transport. Just as before, we find no systematic difference between the position variants (*Figure 3F*) – again suggesting that passage of the PCP through TIM23 is not limiting the overall import rate.

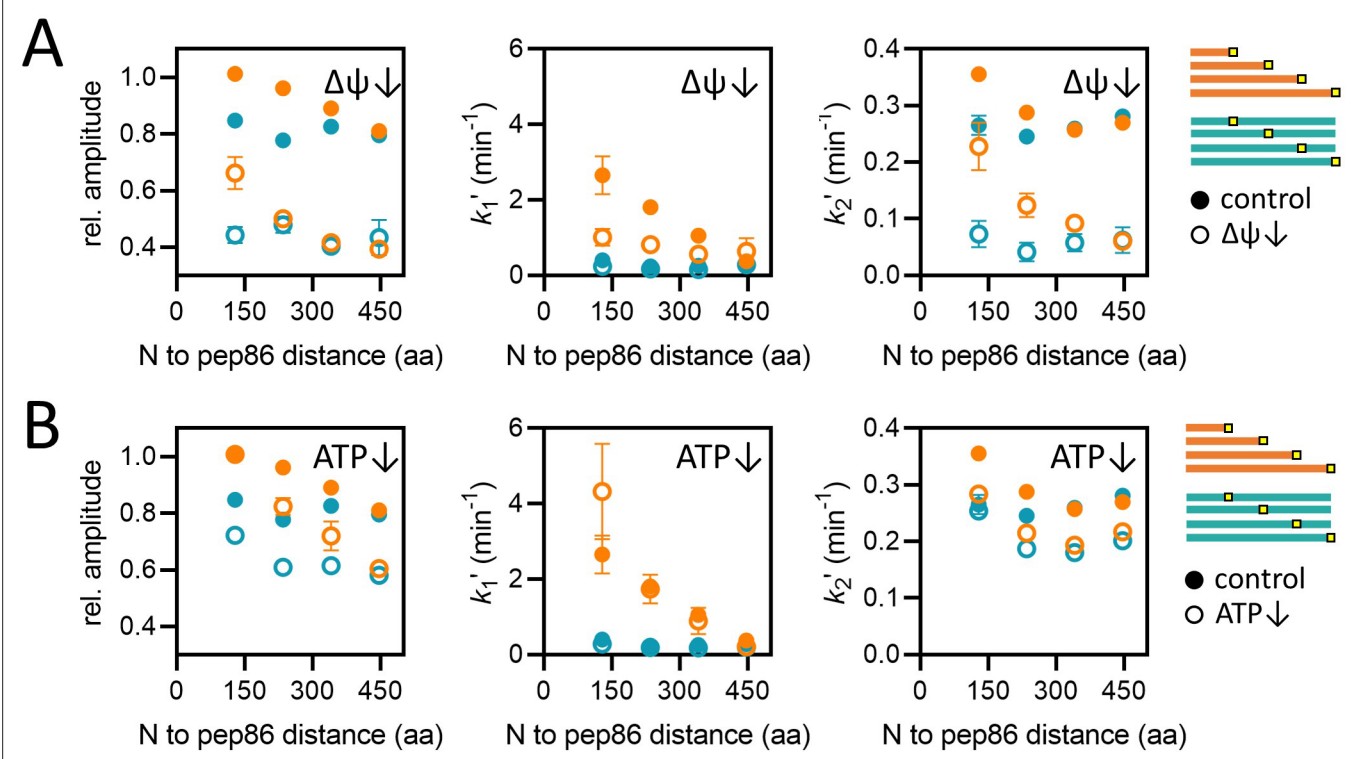

**Figure 4.** Effects of energy depletion on import of the length and position variants. (**A**) Import in the presence (solid circles) or absence (open circles) of $\Delta\phi$, for the length (orange) and position (teal) series. Depletion of $\Delta\phi$ was achieved by a 5-min pre-treatment of mitochondria with 10 nM valinomycin. Plots show amplitude (left), $k_1'$ (middle), and $k_2'$ (right) extracted from two-step fits to import traces as a function of PCP length or pep86 position. Each point is the average and SEM of three independent biological replicates. (**B**) As in panel A, but without (solid circles) or with (open circles) ATP depletion instead of valinomycin. Matrix ATP was depleted by excluding ATP and its regenerating system from the assay mix (see Results section for full description).

The online version of this article includes the following source data and figure supplement(s) for figure 4:

**Source data 1.** Numerical primary (luminescence) and secondary (amplitudes) data corresponding to the graphs in panel A.

**Source data 2.** Numerical primary (luminescence) and secondary (amplitudes) data corresponding to the graphs in panel B.

**Figure supplement 1.** The effect of valinomycin (val) on $\Delta\phi$ and protein import, and confirmation of ATP depletion in the mitochondrial matrix.

**Figure supplement 1—source data 1.** Numerical data corresponding to the graphs in panel A.

**Figure supplement 1—source data 2.** Primary (luminescence) and secondary (amplitude) data corresponding to the graphs in panel B.

**Figure supplement 1—source data 3.** Numerical luminescence data corresponding to the traces shown in panel C.

## Depletion of Δψ and ATP have distinct effects on import

The two driving forces ($\Delta\phi$ and ATP) act at different stages of import (**Figure 1A**), so to help assign $k_1'$ and $k_2'$ we depleted each and measured import of the length and position variants. Mitochondria were pre-treated with valinomycin, a potassium ionophore, which depletes membrane potential in the presence of K[+] ions: 10 nM of valinomycin was used – sufficient to substantially reduce $\Delta\phi$ (**Figure 4—figure supplement 1A**) without completely abolishing precursor transport (**Figure 4—figure supplement 1B**). The effect on the signal amplitude is roughly the same for all length and position variants (**Figure 4A**, left panel) – in contrast to both apparent rate constants: $k_1'$ is somewhat slowed for shorter proteins but largely unaffected for longer ones (**Figure 4A**, middle panel), while $k_2'$ is somewhat slowed for short proteins but dramatically reduced for longer ones (**Figure 4A**, right panel).

Depletion of matrix ATP was achieved simply by excluding ATP and its regenerating system from the assay buffer. Endogenous matrix ATP under these conditions is minimal, as is evident from the fact that import becomes highly sensitive to antimycin A, an inhibitor of oxidative phosphorylation (**Figure 4—figure supplement 1C**). This sensitivity arises because ATP is required for hydrolysis by the ATP synthase to maintain $\Delta\phi$ in the absence of oxidative phosphorylation (**Campanella et al., 2008**).

Import experiments performed with depleted ATP show reduced amplitude, but unlike the response to valinomycin, this effect is more pronounced for the longer PCPs (*Figure 4B*, left panel) – consistent with the proposed role for ATP in promoting transport of the mature part of the PCP. ATP depletion has little or no effect on $k_1'$ (*Figure 4B*, middle panel) and a relatively minor effect on $k_2'$ (*Figure 4B*, right panel), affecting both the length and position variants roughly equally. The minor effect might reflect the low $K_M$ (high affinity) for ATP so that residual concentrations are sufficient to drive transport. Or it could be that ATP hydrolysis increases the rate of a step that is very fast regardless.

## A simple working model for import based on the results above

Taking all the above observations together, we can, as alluded to earlier, propose a simplified model for import that incorporates two major rate-limiting steps. Based on its dependence on PCP concentration (*Figure 3E*; *Figure 3—figure supplement 1*), we assign $k_1'$ to the initial interaction between PCP and TOM. However, the concentration dependence of this step saturates with an apparent $K_M$ of around 100–200 nM. Such saturating behaviour suggests a rapid binding equilibrium followed by a slower step (just as in Michaelis-Menten kinetics), i.e.,

$$PCP \; + \; TOM \; \underset{k_{off}}{\overset{k_{on}}{\rightleftarrows}} \; TOM.PCP \; \overset{k_1}{\rightarrow} \; TOM.PCP^*$$

where * denotes the completed reaction (see below). The strong dependence of $k_1'$ on PCP length (*Figure 3C*, middle panel) provides a clue as to the nature of $k_1$ – it is likely to correspond to passage of the PCP across the OMM, through the TOM complex. The non-linear dependence of step time (1/$k_1'$) on PCP length (*Figure 3D*) also suggests that this step is at least partially diffusional rather than driven by an active energy-dependent directional motor. Furthermore, it suggests that, under these experimental conditions at least, the entire PCP passes through TOM before transport through TIM23 is initiated.

The second rate constant, $k_2'$ is somewhat sensitive to ATP (*Figure 4B*, right panel) and so most likely comes at the end of import, as the contribution of matrix Hsp70 requires at least some of the PCP to be in the matrix. Since $k_2'$ shows very little dependence on PCP length in energised mitochondria (*Figure 3C*, right panel), we propose that it is primarily the $\Delta \psi$-dependent insertion of the pre-sequence through TIM23, not the subsequent passage of the unfolded passenger domain that is limiting (although both presumably contribute to the apparent rate constant). However, under conditions of $\Delta \psi$ depletion, a length-dependence of $k_2'$ emerges (*Figure 4A*, right panel, orange open circles). This suggests that $\Delta \psi$ drives the transport of the mature domain of the precursor, and not exclusively the MTS, and is consistent with the import rate of the rest of the PCP being affected by ($\Delta \psi$ [*Schendzielorz et al., 2017*], and see also below). It is also possible that transport of longer PCPs has a higher chance of failure, with the PCP slipping back into the IMS – this would be a useful mechanism to prevent TIM23 complexes becoming blocked with mis-folded/compacted PCPs and would explain the difference in the effect of $\Delta \psi$ depletion on the length and position variants.

Putting all of this together, we propose the following minimal kinetic scheme for PCP import:

$$PCP_{out} \; + \; TOM \; \underset{k_{off}}{\overset{k_{on}}{\rightleftarrows}} \; TOM.PCP_{out} \; \overset{k_1}{\rightarrow} \; TOM.PCP_{IMS} \; \overset{k_2}{\rightarrow} \; PCP_{in},$$

where the subscript to PCP indicates its location (<u>out</u>side the OMM, in the <u>IMS</u>, or <u>in</u>side the matrix). In this model, $k_{on}$ and $k_{off}$ are both fast compared with $k_1$, and give an affinity ($K_d = k_{off}/k_{on}$) of the order of 100 nM, similar to the affinity of a bacterial secretion pre-proteins to bacterial inner membrane vesicles (*Hartl et al., 1990*). The two extracted rate constants can be approximately determined as ([PCP] designates PCP concentration):

$$k_1' \; \sim k_1 \frac{[PCP]}{K_d + [PCP]} \; and \; k_2' \; \sim \; k_2$$

This model fits the data, and we believe it is the most reasonable interpretation of the above experimental results. However, it still leaves open several questions, notably the extent to which $k_1$ and $k_2$ are reversible. For example, the fact that $k_1'$ is somewhat affected by valinomycin (*Figure 4A*, middle panel) suggests that $k_1$ is reversible. Given that passage through TOM can occur in the absence of $\Delta \psi$ (*Mayer et al., 1993*; *Lill et al., 1992*), slowing $k_2$ would then leave more opportunity for diffusion back

out of the IMS through TOM, a process that occurs in the absence of ATP (*Ungermann et al., 1996*). In addition, we cannot determine from this data exactly at what stage handover from TOM to TIM23 occurs. The results suggest that PCP passes through TOM completely before engaging with TIM23, but it is not clear whether this is a necessary part of the mechanism or merely an effect of the relative rates under these conditions. Nor can we determine whether handover from TOM to TIM23 is direct or if the PCP can dissociate from TOM before binding to TIM23.

### Removal of the OMM selectively affects $k_1'$ but not $k_2'$

To confirm our assignments of $k_1'$ and $k_2'$, we tested the effect of removing the OMM (producing mitoplasts) on import kinetics. If the model is correct, this should eliminate steps at the OMM (i.e. $k_1$) but not at the IMM ($k_2$). Mitoplasts were isolated from yeast mitochondria using an optimised concentration of 2 mg/ml digitonin, and successful removal of the OMM was confirmed by respirometry (*Figure 5—figure supplement 1A, B*). Addition of ADP to mitoplasts stimulates oxygen consumption less than in mitochondria, indicating a lower PMF – since PMF limits oxygen consumption in the

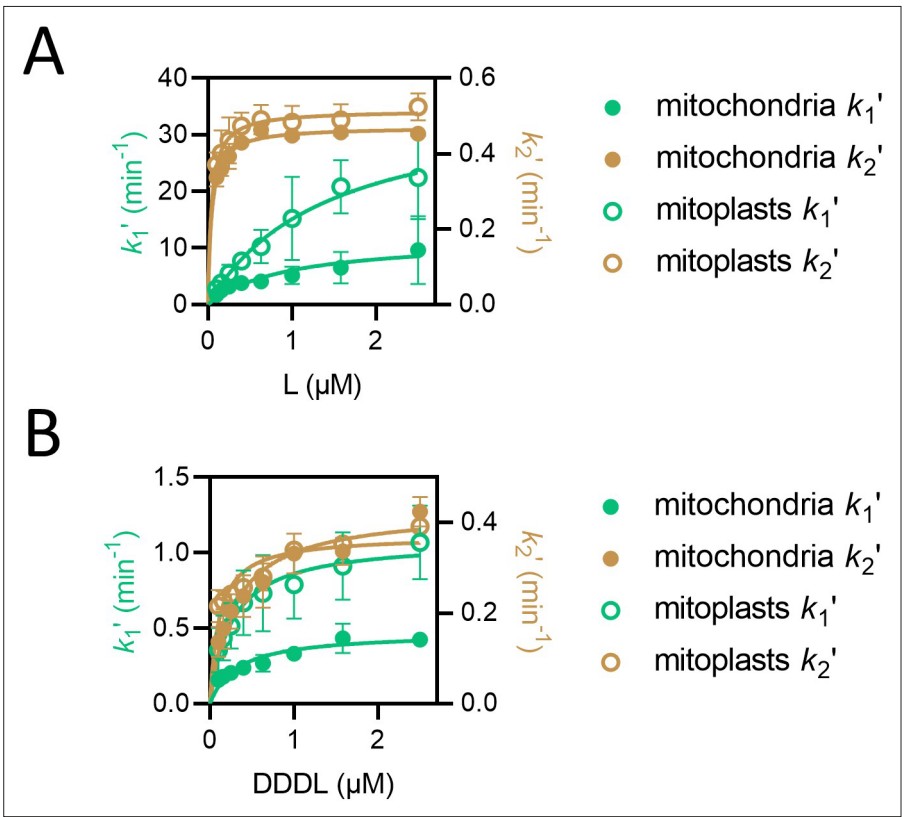

**Figure 5.** $k_1'$ but not $k_2'$ is affected by removal of the outer mitochondrial membrane. (**A**) Values of $k_1'$ (teal symbols) and $k_2'$ (brown symbols) from fitting of the two-step model to luminescence data from import of L (PCP-pep86) at concentrations ranging from 0.1 to 2.5 µM into mitochondria (solid circles) and mitoplasts (open circles). Data are shown as the mean of three biological repeats with error bars showing SEM. Data were fit to the Michaelis-Menten equation, and resulting fits are shown in respective colours. (**B**) As in panel A but with DDDL instead of L.

The online version of this article includes the following source data and figure supplement(s) for figure 5:

**Figure supplement 1.** Validation of outer mitochondrial membrane removal in mitoplast preparations, and a comparison of pre-sequence-containing precursor import kinetics in mitochondria and mitoplasts.

**Figure supplement 1—source data 1.** Numerical data corresponding to the traces shown in panel A, and data from all biological repeats of the experiment from which the data in panel B was derived.

**Figure supplement 1—source data 2.** Numerical primary (luminescence) and secondary data corresponding to the graph in panel A (and the graph on the left in *Figure 5—figure supplement 1C*).

**Figure supplement 1—source data 3.** Numerical primary (luminescence) and secondary ($k_1'$ and $k_2'$) data corresponding to the graph in panel B (and the graph on the right in *Figure 5—figure supplement 1C*).

absence of ADP (*Figure 5—figure supplement 1A, B*; *Chance and Williams, 1955*). However, much of this respiration is recovered by the addition of exogenous cytochrome c to mitoplasts (but not mitochondria) due to the loss of endogenous cytochrome c and accessibility of the IMM (*Figure 5—figure supplement 1A, B*). Taken together, this confirms that the mitoplast treatment employed has effectively removed the OMM, without too much damage to the IMM.

We next compared the kinetics of import of the shortest and longest versions of the length variant proteins, at various concentrations, between intact mitochondria and mitoplasts. Even without the OMM, a two-step model is required to describe the data; however, while the values for $k_2'$ are similar in mitoplasts to those observed in mitochondria (*Figure 5*), $k_1'$ is substantially increased by removal of the OMM for both the shortest (*Figure 5A*) and longest (*Figure 5B*) of the length variant PCPs. The fact that $k_2'$ is unaffected by removal of the OMM confirms that it takes place at the IMM, as predicted. The faster $k_1'$ presumably corresponds to a new process – most likely direct association of the PCP with the IMM. For the shortest protein, L, signal amplitude is almost completely unaffected by OMM removal, while for the longest, DDDL, amplitude is only moderately reduced (*Figure 5—figure supplement 1C*). Most likely, the latter reduction is due to a slightly depleted $\Delta\phi$ in mitoplasts.

## Contrasting effects of PCP net charge on the amplitude and rate of import

$\Delta\phi$, the electrical component of the PMF (positive outside), has been proposed to act primarily upon positively charged residues in the PCP, pulling them through electrophoretically (*Martin et al., 1991*; *Geissler et al., 2000*; *Truscott et al., 2001*). To test this idea, we designed a series of proteins, based on an engineered version of a classical import substrate: the N-terminal section of yeast cytochrome $b_2$ lacking the stop-transfer signal (Δ43–65) to enable complete matrix import (*Gold et al., 2014*). The variant PCPs differed only in the numbers of charged residues *Figure 6A*; identical in length (203 amino acids), but spanning 5.43 units of pI ranging from 4.97 to 10.4. Import of these charge variants under saturating conditions (1 µM PCP) was measured using the NanoLuc assay as above and representative traces are shown in *Figure 6B* (with complete data in *Figure 6—figure supplement 1*).

The most immediately striking observation is that amplitude is strongly inversely correlated with net charge of the PCP – the opposite of what might be expected given the direction of $\Delta\phi$ (inside negative; *Figure 6C*). To understand why this would be, we turned to our earlier interpretation of signal amplitude: that it is limited by the availability of $\Delta\phi$. If transport of positively charged residues consumes $\Delta\phi$ while transport of negatively charged residues replenishes or maintains it, this could explain why negatively charged proteins accumulate to a higher level.

To test this hypothesis, we monitored $\Delta\phi$ in isolated mitochondria over time by measuring tetramethylrhodamine methyl ester (TMRM) fluorescence, then assessed the effect of adding the PCPs with differing net charge (*Figure 6D*). The PCPs did indeed cause strong depolarisation of $\Delta\phi$, and moreover, this effect diminished with increasing net negative charge. Increasing net positive charge (above zero) did not seem to result in enhanced depletion of $\Delta\phi$, but TMRM does not resolve $\Delta\phi$ well in this range (*Figure 4—figure supplement 1A*), so this does not necessarily mean that this effect is not occurring. Note that this depletion of $\Delta\phi$ is not caused by NanoLuc activity: there is no Nano-Glo luciferase assay substrate present in this assay, and a similar reduction of $\Delta\phi$ is observed even for PCPs lacking pep86 (*Figure 6—figure supplement 2A*).

A second prediction from the above hypothesis is that membrane depolarisation prior to protein import will abolish the correlation between net charge and amplitude. This is indeed exactly what we observe: pre-treatment of mitochondria with valinomycin reduces amplitudes for all PCPs, but the effect is greater for more negatively charged PCPs, bringing all amplitudes to about the same level (*Figure 6E*). Depleting ATP, meanwhile, has very little effect on amplitude, just as for the Acp1-based PCPs. Therefore, the counterintuitive increase in the import yield of more negative PCPs is an indirect effect of diminishing substrate-induced membrane depolarisation.

The response of import rates to PCP charge is in direct contrast to the amplitude. It is clear from looking directly at the import traces that positively charged PCPs are imported much faster than negatively charged ones (albeit reaching a lower final amplitude; *Figure 6B–C*). This is consistent with $\Delta\phi$ specifically assisting the transport of positively charged residues (*Martin et al., 1991*; *Geissler et al., 2000*; *Truscott et al., 2001*). Furthermore, the more positively charged PCPs do not show the characteristic delay before signal appears that indicates a two-step transport model (see *Figure 1C*);

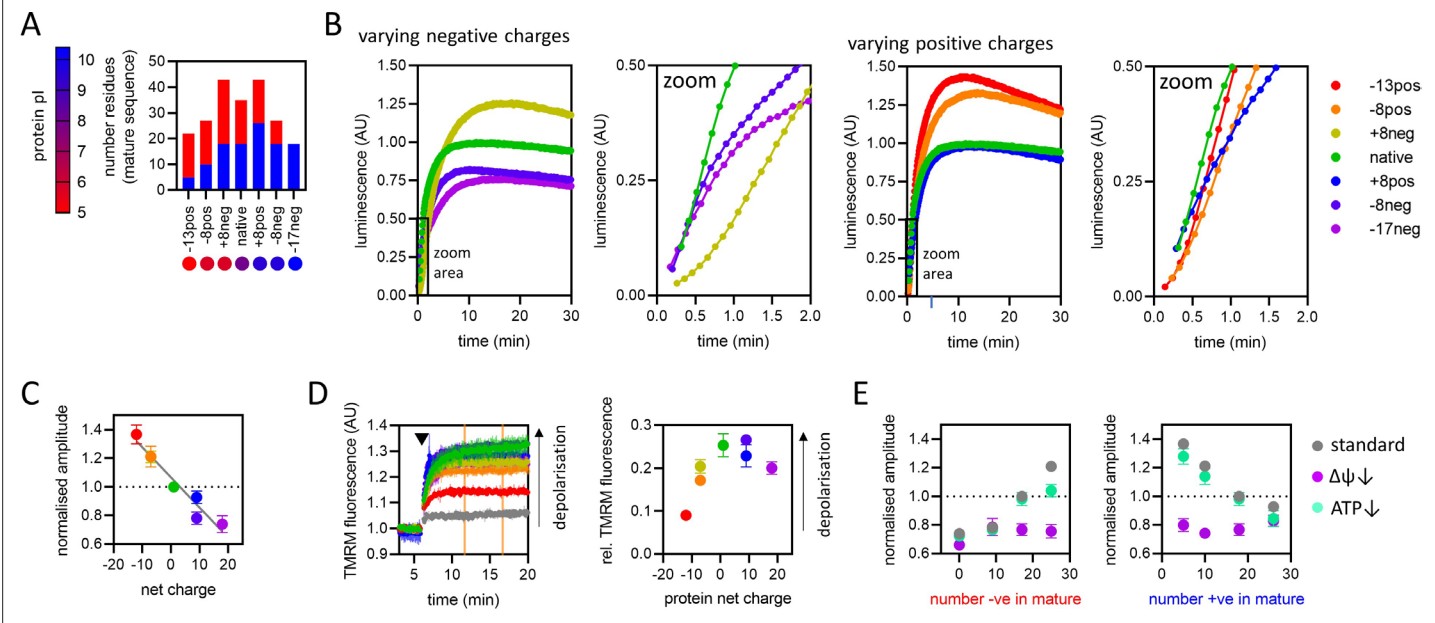

**Figure 6.** The effect of pre-sequence-containing precursor (PCP) charge on import kinetics. (**A**) Overview of the charge variant protein series, showing numbers of positively (blue) and negatively (red) charged residues in the mature part of each protein, and symbols for each protein with colours corresponding to theoretical pI, according to the scale shown on the left. All proteins in the charge variant series have the same length (203 amino acids) and are based on the N-terminal section of yeast cytochrome $b_2$ lacking the stop-transfer signal (Δ43–65) to enable complete matrix import (**Gold et al., 2014**). (**B**) Import traces for the charge variant proteins in which the number of negative (left) and positive (right) charges is varied, normalised to the native PCP, coloured by rainbow from most negative (red) to most positive (violet). Data shown are a single representative trace; this is because starting points for each data set are slightly offset due to the injection time of the plate reader. Full data – three biological replicates each performed in duplicate – are shown in **Figure 6—figure supplement 1**. (**C**) Amplitudes obtained from panel B as a function of net charge (coloured as in panel B), with a line of best fit shown. The data point for the +8neg protein (yellow) is in the same position as the –8pos protein (orange) and is mostly hidden. Data are the mean ± SD of three biological replicates each performed in duplicate. (**D**) Tetramethylrhodamine methyl ester (TMRM) fluorescence over time in isolated yeast mitochondria (left), with PCPs added at the time indicated by arrowhead. A no protein control (buffer only) is shown in grey, and the remaining traces are shown with the PCP coloured as in panel B. Average TMRM fluorescence over a 5-min window (between orange vertical lines) was calculated for each trace then plotted, relative to no protein control, against protein net charge (right). Data shown is mean ± SD from three biological repeats. (**E**) Amplitude (normalised to the native PCP in standard conditions) of import signal for the charge variants, where number of negatively (left) or positively (right) charged residues is varied, under standard reaction conditions (grey) or when Δφ (purple) or ATP (green) is depleted. Each data point is the mean ± SEM from three biological repeats (shown in **Figure 6—figure supplement 1B, C**). Error bars smaller than symbols are not shown.

The online version of this article includes the following source data and figure supplement(s) for figure 6:

**Source data 1.** Numerical data corresponding to the bar chart in panel A.

**Source data 2.** Numerical data corresponding to the luminescence traces in panel B.

**Source data 3.** Numerical data corresponding to the graph in panel C.

**Source data 4.** Numerical data corresponding to the graphs in panel D.

**Source data 5.** Numerical data corresponding to the graphs in panel E.

**Figure supplement 1.** Complete import traces for the data in **Figure 6**.

**Figure supplement 1—source data 1.** Numerical luminescence data corresponding to the traces in panels A–C.

**Figure supplement 2.** Dissipation of Δφ with protein import is not an artefact specific to the NanoLuc assay or of the method of Δφ measurement.

**Figure supplement 2—source data 1.** Numerical data corresponding to the TMRM data.

in this case a simpler single-step model is sufficient. This means that one of the two steps ($k_1$ or $k_2$) has become much faster, such that the other is completely rate limiting. Either positively charged PCPs cross the OMM much faster than negative ones, or they engage with TIM23 much faster (**Figure 6B**); both of these possibilities seem feasible. There is even an indication, particularly for the most positive PCP, of a rapid 'burst' of import followed by a slower phase (**Figure 6B**, green, blue, indigo and violet traces in the zoomed panels). This might suggest multiple import events per TIM23, with the first

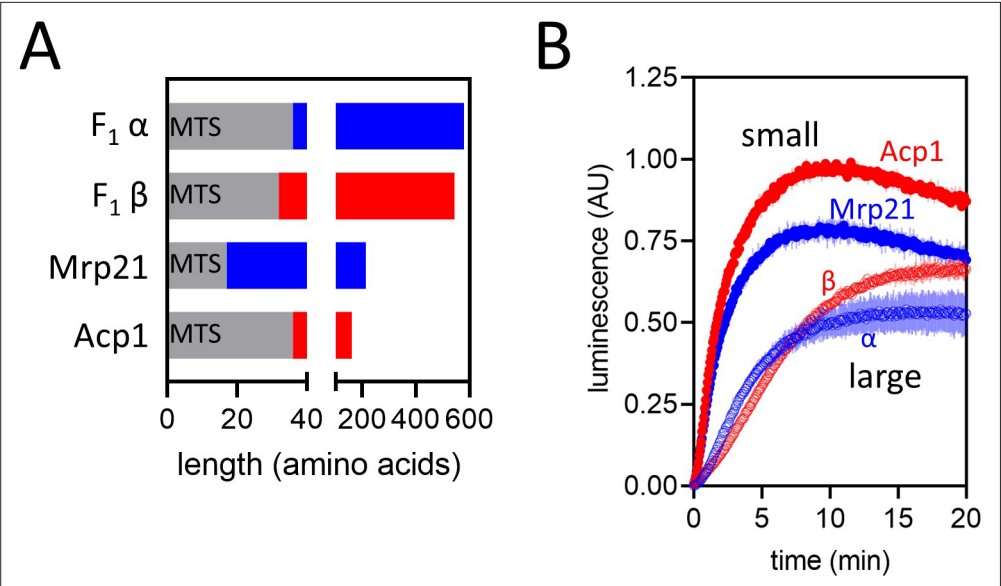

**Figure 7.** Import of pep86-fused native precursors. (**A**) Schematic representation of four native pre-sequence-containing precursors (PCPs) : F$_1\alpha$ (long, positively charged, predicted pI of mature part is 6.98), F$_1\beta$ (long, negatively charged, predicted pI of mature part is 5.43), Mrp21 (short, positively charged, predicted pI of mature part is 10.00), and Acp1 (short, negatively charged, predicted pI of mature part is 4.87). (**B**) Import traces for the four PCPs in panel A under standard conditions (1 μM PCP), normalised to Acp1. Each trace is the mean ± SD of three biological repeats.

The online version of this article includes the following source data for figure 7:

**Source data 1.** Numerical data corresponding to the bar chart in panel A.

**Source data 2.** Numerical data corresponding to the luminescence traces in panel B.

one very fast and subsequent ones limited by a slower process, e.g., resetting of TIM23. Designing a set of PCPs and mathematical model to test this could be a useful future approach to dissecting the transport process in more detail.

## Validation of the observed charge and length effects with native PCPs

While the use of artificial PCPs, as above, allows their properties to be varied in a systematic manner, it is possible that these modifications will affect native features with fundamental roles in the import process. To confirm that the above observations hold true for native PCPs, we performed import experiments with four pep86-tagged native PCPs differing in length and charge. We chose the F$_1$ α and β subunits of the mitochondrial ATP synthase, both large proteins (>500 amino acids) with mature amino acid sequences differing in predicted pI by ~1.55 (F$_1$ $\beta$=5.43 and F$_1$ $\alpha$=6.98); and two smaller proteins (<200 amino acids), Acp1 and Mrp21, with predicted mature sequence pIs of 4.87 and 10.00, respectively (*Figure 7A*).

Consistent with our earlier results, we see higher amplitudes for the shorter and more negatively charged PCPs (*Figure 7B*), and faster import of the shorter PCPs than the longer ones (*Figure 7B*). The effect of net charge seen above – faster rates for positively charged PCPs – holds true for the larger PCPs (*Figure 7B*). The small PCPs do not appear to differ significantly in import rate (*Figure 7B*); perhaps for short proteins, the effect of the mature domain charge is overwhelmed by the presence of the highly positively charged MTS. Overall, these results suggest that conclusions drawn from data collected with artificial PCPs are applicable for native ones as well.

## Discussion

Protein import into mitochondria is, by nature, a complicated process with machineries in two membranes having to coordinate with one another as well as with parallel import pathways to deliver a wide range of proteins to their correct destinations. Here, we have built a minimal mechanistic

model of one of the major import routes – the TOM-TIM23$^{MOTOR}$ pathway of matrix proteins – using a high-resolution import assay based on NanoLuc (*Pereira et al., 2019*). Our results suggest that two major distinct events are responsible for the majority of the PCP transit time: passage of the PCP through the TOM complex and insertion of the pre-sequence through the TIM23$^{MOTOR}$ complex. By contrast, the initial binding of PCP to TOM is fairly rapid, as is passage of the mature PCP domain through TIM23. Crucially, the rates of the different steps correlate very poorly with the amount of PCP in the matrix when the reaction ends, which has always been the conventional readout of import. Therefore, this pre-steady-state kinetic approach has been critical for this study and will be for further dissection of PCP transport through the TOM and TIM23$^{MOTOR}$ complexes and additional pathways that collectively comprise the mitochondrial protein import machinery.

Import appears to be largely single turnover under our experimental conditions, i.e., each import site only imports a single PCP. While this is fortuitous in that it allows us to access pre-steady-state events easily, it is incongruent with mitochondrial protein import *in vivo*. Nonetheless, this almost certainly holds true for decades of experiments using the classic method and offers an explanation as to why these methods require such high concentrations of mitochondria for detection of import. We propose that, under experimental conditions, import is limited by the amount of energy available in the form of $\Delta\phi$. Indeed, measurements of $\Delta\phi$ using TMRM confirm that PCP import causes a depolarisation of the IMM that is not restored. Also consistent with $\Delta\phi$ being consumed, we find that the PCPs that require more total energy to import (such as longer ones) or that are likely to consume more $\Delta\phi$ (positively charged ones) reach a lower final concentration in the mitochondrial matrix at saturating PCP concentrations.

Importantly, we were able to show that the IMM depolarisation is an effect of import rather than an inherent problem with the mitochondria preparation. The isolated mitochondria are indeed capable of building a PMF via respiration (*Figure 5—figure supplement 1A* and *Figure 6—figure supplement 2*), and furthermore, recovering PMF after dissipation caused by externally added ADP (*Figure 6—figure supplement 2B*). Therefore, the sustained dissipation of $\Delta\phi$ must be a consequence of the import process due to charge associated with the PCP itself and/or co-transporting ions. It is certainly conceivable that the acute addition of saturating quantities of PCP to isolated mitochondria, deployed here, overwhelms their ability to recover $\Delta\phi$. The demand on the import apparatus *in vivo* would be lower. With sub-saturating levels of precursor proteins, mitochondria inside healthy cells would more easily recover their $\Delta\phi$, enabling multiple turnovers of PCP import.

The mechanism by which $\Delta\phi$-depletion leads to single turnover conditions may relate to the requirement of $\Delta\phi$ for dimerisation of TIM23 and recruitment of Tim44; both needed for PCP delivery to the matrix (*Bauer et al., 1996*; *Martinez-Caballero et al., 2007*; *Demishtein-Zohary et al., 2017*; *Ting et al., 2017*; *Ramesh et al., 2016*). PCPs bind only to TIM23 complexes containing two Tim23 subunits and during transport this dimeric organisation becomes disrupted (*Bauer et al., 1996*). Therefore, PCP-induced loss of $\Delta\phi$ in isolated mitochondria would prohibit the resetting of the TIM23 complex required for further turnovers. Alternatively, the redistribution of charge that occurs with PCP transport could cause localised patches of charge that impede further import events until redistributed. Another possible explanation is that the import process itself compromises the integrity of the TIM23 channel and thereby results in concomitant dissipation of $\Delta\phi$. This re-priming event could provide an interesting area for future studies. Indeed, the failure of import re-initiation may not just be a quirk of compromised mitochondria in isolation. When the bioenergetic fitness of mitochondria *in situ* declines, then the failure of TIM23 turnover would bring about an increase in mistargeted PCPs and have profound implications for the cell – the activation of the mitochondrial unfolded protein response, as previously described (*Wrobel et al., 2015*). Therefore, knowledge of the molecular basis of the switch between single and multi-turnover behaviour could prove to be very important for understanding mitochondrial health and disease.

The transfer of PCPs from TOM to TIM23 is thought to involve cooperative interactions of subunits of the two complexes (*Gomkale et al., 2021*; *Callegari et al., 2020*). But the extent to which transport of PCPs across the OMM and IMM is coupled *in vivo* remains unknown. It has been suggested that the rate of PCP passage through the OMM is one factor that determines whether PCPs are transferred to the matrix or released laterally into the IMM (*Harner et al., 2011b*), implying simultaneous and cooperative activities of TOM and TIM23. PCPs have been captured spanning both membrane complexes at the same time in super-complexes of ~600 kDa (*Gomkale et al., 2021*; *Dekker et al.,*

*1997*; *Gold et al., 2014*; *Chacinska et al., 2010*), suggesting that import through TOM does not have to be complete before import through TIM23 can begin.

Contrasting with this, however, there is also evidence to suggest that the TOM and TIM23 complexes can transport PCPs independently in steps that are not necessarily concurrent. The *in vivo* existence of TOM-TIM23 super-complexes is unconfirmed; they have been detected only when engineered PCPs with C-terminal domains that cannot pass through TOM are used (*Chacinska et al., 2003*), and only under these artificial conditions do TOM and TIM23 subunits co-immuno-precipitate or co-migrate on native polyacrylamide gels (*Horst et al., 1995*). Perhaps their assembly is more dynamic and transient, relying on other OMM-IMM contact sites such as the MICOS complex (*von der Malsburg et al., 2011*; *Hoppins et al., 2011*; *Harner et al., 2011a*). Moreover, the N-terminal domain of Tim23, which tethers the IMM and OMM, is not required for either PCP import though TIM23, or TOM-TIM23 super-complex formation (*Chacinska et al., 2003*).

Our results confirm that direct handover from TOM to TIM23 is not absolutely required for import. Import of proteins into mitoplasts, observed here and previously (*Hwang et al., 1989*; *Ohba and Schatz, 1987*; *van Loon and Schatz, 1987*), shows that TOM and soluble factors in the IMS are dispensable for import through TIM23. Additionally, the rate of the import step that we assigned to insertion of precursor into the TIM23 complex ($k_2'$) remained relatively unchanged by removal of the OMM – unlikely to be the case if the mechanistic process was significantly altered (e.g. by loss of coupled import directly from TOM). And furthermore, we see some PCP concentration dependence of $k_2'$; if direct interaction of TOM with TIM23 were strictly required then $k_2$ would not be affected by PCP concentration, but if PCP can accumulate in the IMS this would explain our finding.

The data here also suggest that transport of a PCP through TOM is reversible and therefore possible in the absence of TIM23 activity. This could serve as a checkpoint to ensure that the TIM23 channel does not get blocked when transport across the IMM is compromised. Retro-translocation through TOM to cytosolic proteasomes would be an efficient system for clearance of proteins from the IMS, which could even serve as a protein reservoir for quality control purposes, where pathological protein accumulation can be easily monitored. Reverse transport of proteins through TOM, and in some cases also through TIM23, has been observed previously, although this process is not well understood. For example, proteins that are reduced or conformationally unstable in the IMS can retro-translocate to the cytosol via TOM40, and the efficiency of this process is relative to protein size (both linear length and 3D complexity); smaller proteins are more efficiently retro-translocated (*Bragoszewski et al., 2015*). Notably, under physiological conditions, PINK1 is cleaved in the IMM by PARL, releasing the C-terminal region for release back to the cytosol for proteosomal degradation. However, the process is not well understood, such as if, and how, it is regulated, and if an energetic driving force is required.

Despite all this, the fact that coupling of TOM and TIM23 import is not required does not exclude the possibility that it happens *in vivo*. Import experiments are performed by adding a large excess of a single PCP to mitochondria that presumably have all their TOM sites unoccupied. This could potentially cause a flood of PCP to enter through TOM – which is in excess of TIM23 (*Sirrenberg et al., 1997*; *Dekker et al., 1997*) – and overwhelm the ability of the coupled system to cope, leading to a build-up in the IMS. The high sensitivity of the NanoLuc system could potentially allow future experiments under more native-like conditions, in which PCPs containing pep86 are provided alongside other, unlabelled TOM substrates – perhaps also with other cytosolic components such as chaperones.

Overall, the above analysis provides good estimates of the two rate-limiting steps for import and provides evidence as to the constraints that act upon the other (non-rate-limiting) steps. The model presented provides us with a foundation for the development of a complete kinetic model for mitochondrial import, as has been recently achieved for the bacterial Sec system (*Allen et al., 2020*). Already the comparison of the two systems has proven to be very insightful: where, in bacteria, transport initiation is rapid with length-dependent passage of the mature domain rate-limiting (*Allen et al., 2020*), in mitochondria this is reversed, with slower initiation then rapid processive transport. It is likely that these differences reflect distinct selection pressures on the two transport systems. For bacteria, the rate of secretion is probably one factor limiting their ability to divide, so its optimisation will have a direct bearing on their competitiveness. Whereas in mitochondria, it is more critical to avoid mistargeting proteins to the matrix; hence the step where proteins are committed to matrix import is slower and more controlled.

## Materials and methods

### Strains and plasmids

*Escherichia coli* α-select cells were used for amplifying plasmid DNA and BL21 (DE3) used for protein expression. Genes encoding pep86 (trademarked as 'SmBiT' *Dixon et al., 2016*)-tagged mitochondrial PCP proteins (from MWG Eurofins or Thermo Fisher Scientific) were cloned into either pBAD, pRSFDuet, or pE-SUMOpro. YPH499 yeast cell clones transformed with pYES2 containing the mt-11S gene under control of the GAL promoter, used previously (*Pereira et al., 2019*), were used for isolation of mitochondria containing matrix-localised 11S (trademarked as 'LgBiT' *Dixon et al., 2016*). *E. coli* cells were routinely grown at 37°C on Luria Broth (LB) agar and in either LB or 2xYT medium containing appropriate antibiotics for selection. Yeast cells were grown at 30°C on synthetic complete dropout (Formedium) agar supplemented with 2% glucose, penicillin, and streptomycin, or in synthetic complete dropout medium, supplemented with 3% glycerol, penicillin, and streptomycin in baffled flasks. For yeast cells with mitochondrial matrix-localised 11S, mt-11S was expressed by adding 1% galactose at mid-log phase, 16 hr prior to harvesting of cells.

### Protein production and purification

BL21 (DE3) cells from a single colony, containing the chosen protein expression plasmid, were grown in LB overnight then sub-cultured in 2×YT medium until $OD_{600}$ reached 0.6. For pBAD and pRSFDuet plasmids, protein expression was induced by adding arabinose or isopropyl ß-D-1-thiogalactopyranoside (IPTG), respectively. Cells were harvested 2–3 hr later and lysed using a cell disrupter (Constant Systems Ltd.). Proteins were purified from inclusion bodies using Nickel affinity chromatography on prepacked HisTrap FF columns (Cytiva, UK), followed by ion exchange chromatography on either HiTrap Q HP or HiTrap SP HP columns (Cytiva, UK) depending on protein charge, described in full previously (*Pereira et al., 2019*). Proteins from pE-SUMOpro plasmids (those containing DHFR domains) were expressed by adding IPTG, and cells harvested after 18 hr of further growth at 18°C. Proteins were purified at 4°C from the soluble fraction, essentially as before (*van Loon and Schatz, 1987*), but with 250 mM NaCl in their 'Buffer C'. A further purification step on a HiLoad 16/60 Superdex gel filtration column (Cytiva, UK) was included to remove remaining contaminants. A full list of PCPs, their amino acid sequences, and respective expression vectors are given in *Supplementary file 1*.

### Isolation of mitochondria from yeast cells

Yeast cells were harvested by centrifugation (4000× *g*, 10 min, room temperature) and mitochondria isolated by differential centrifugation (*Daum et al., 1982*). Briefly, cell walls were digested with zymolyase in phosphate-buffered sorbitol (1.2 M sorbitol, 20 mM potassium phosphate pH 7.4), after being reduced with dithiothreitol (DTT; 1 mM DTT in 100 mM Tris-SO4 at pH 9.4, for 15 min at 30°C). Cells were disrupted at 4°C with a glass Potter-Elvehjem homogeniser with motorised pestle in a standard homogenisation buffer (0.6 M sorbitol, 0.5% (w/v) BSA, 1 mM phenylmethylsulfonyl fluoride (PMSF), 10 mM Tris-HCl pH 7.4). The suspension was centrifuged at low speed (1480× *g*, 5 min) to pellet unbroken cells, cell debris, and nuclei, and mitochondria harvested from the supernatant by centrifugation at 17,370× *g*. The pellet, containing mitochondria, was washed in SM buffer (250 mM sucrose and 10 mM MOPS, pH 7.2) and then centrifuged at low speed again to remove remaining contaminants. The final mitochondrial sample, isolated from the supernatant by centrifugation (17,370× *g*, 15 min), was resuspended in SM buffer and protein quantified by bicinchoninic acid assay (*Smith et al., 1985*) using a bovine serum albumin protein standard. Mitochondria were stored at –80°C, at a concentration of 30 mg/ml in single-use aliquots, after being snap frozen in liquid nitrogen. Biological replicates were obtained using mitochondria isolated from independent cultures of yeast.

### Preparation of mitoplasts from isolated yeast mitochondria

Mitoplasts were prepared at 4°C by incubation of mitochondria in SM buffer containing 2 mg/ml (unless otherwise specified) digitonin at a ratio of 0.2 g per gram of mitochondrial protein. After 15 min, they were centrifuged (12,000× *g*, 5 min), washed in SM buffer, and collected by centrifugation (as before).

## Respirometry with mitochondria and mitoplasts

Respirometry was performed at 25°C with an Oxygraph-2k respirometer (Oroboros Instruments, Austria) using DatLab Version 6.1.0.7 software. In the Oxygraph-2k chambers, mitochondria (or equivalent amounts of mitoplasts) were resuspended in respiration buffer (250 mM sucrose, 80 mM KCl, 1 mM $K_2HPO_4$/$KH_2PO_4$, 5 mM $MgCl_2$, and 10 mM MOPS-KOH) at a concentration of 70 µg/ml (for intact mitochondria). Routine respiration was measured before the following additions were made, each time allowing respiration to stabilise before the next addition: NADH (2 mM), ADP (1 mM), cytochrome c (10 µM), oligomycin (15 µM), and carbonyl cyanide m-chlorophenyl hydrazone (CCCP) (0.5 µM).

## Western blotting

Samples of mitochondria from yeast cells were solubilised in SDS-PAGE sample buffer (2% [w/v] SDS, 10% [v/v] glycerol, 62.5 mM Tris-HCl pH 6.8, 0.01% [w/v] bromophenol blue, and 25 mM DTT), and fractionated on a 15% (w/v) acrylamide, 375 mM Tris pH 8.8, 0.1% (w/v) SDS gel with a 5% (w/v) acrylamide, 126 mM Tris pH 6.8, 0.1% (w/v) SDS stacking gel, in Tris-Glycine running buffer pH 8.3 (25 mM Tris. 192 mM glycine, 0.1% [w/v] SDS). Proteins were electro-transferred to polyvinylidene difluoride (PVDF) membrane in 10 mM $NaHCO_3$, 3 mM $Na_2CO_3$, then membranes incubated in blocking buffer (Tris-buffered saline (TBS) [50 mM Tris-Cl pH 7.5, 150 mM NaCl] containing 0.1% [v/v] Tween 20% and 5% [w/v] skimmed milk powder). 11S protein was detected with a rabbit polyclonal antibody (Promega), Tom40 with a rabbit polyclonal antibody (Cambridge Research Biochemicals, Billingham, UK), and the myc tag with a mouse monoclonal antibody (Cell Signaling Technology). Primary antibody incubations were at 4°C for 18 hr in blocking buffer. Membranes were washed in TBS containing 0.1% (v/v) Tween 20, three times, each for 10 min, before incubation for 1 hr with an horseradish peroxidase-conjugated goat secondary antibody against rabbit IgG (Thermo Fisher Scientific), in blocking buffer. Membranes were washed, as before, and antibodies visualised using 1.25 mM luminol, with 198 µM coumaric acid as enhancer, and 0.015% (v/v) $H_2O_2$ in 100 mM Tris-Cl pH 8.5. Images were acquired on a Odyssey Fc Imager (LI-COR Biosciences) and densitometric analysis performed using Image Studio Software (LI-COR Biosciences).

## NanoLuc import assay

Unless stated otherwise, import experiments were performed at 25°C with mt-11S mitochondria diluted to 50 µg/ml in import buffer (250 mM sucrose, 80 mM KCl, 1 mM $K_2HPO_4$/$KH_2PO_4$, 5 mM $MgCl_2$, 10 mM MOPS-KOH, and 0.1% [v/v] Prionex reagent [Merck], pH 7.2), supplemented with 2 mM NADH, 1 mM ATP, 0.1 mg/ml creatine kinase, 5 mM phosphocreatine, 0.25× Nano-Glo luciferase assay substrate (supplied at 100×, Promega UK), and 1 µM pep86-tagged PCP protein. We also added 10 µM GST-Dark protein; a fusion of glutathione S-transferase and a peptide with high affinity for 11S that inhibits pep86 binding and concomitant enzymatic activity and thereby reduces background signal caused by trace amounts of 11S outside the mitochondrial matrix (*Pereira et al., 2019*). Mitochondria and GST-Dark were added to 1× import buffer at 1.25× final concentrations (mixture 1), and pep86-tagged PCP, NADH, ATP, creatine kinase, and phosphocreatine added to 1× import buffer at 5× final concentrations (mixture 2) so that import reactions could be started by the injection of 4 vols mixture 1 onto 1 vol mixture 2. For experiments that involved MTX, PCPs were incubated in the presence of 5.57 mM DTT and in the presence or absence of 524 µM MTX and 524 µM NADPH (15 min at 21°C). Urea was added for a final concentration of 3.5 M, 10 min before addition to the import mixture (as 4 µl at 1.25 µM). Final concentrations of MTX and NADPH were 5 µM. For measurement of pep86 binding to 11S in solution, mitochondria were first solubilised by incubation with digitonin (5 mg/ml) at 4°C for 15 min. In selected experiments, depletion of $\Delta\psi$ was achieved by pre-treating mitochondria for 5 min with 10 nM valinomycin, and depletion of ATP was achieved by omitting ATP, creatine kinase, and phosphocreatine from the reaction. ATP depletion was verified by monitoring sensitivity of mitochondria to a 5 min pre-treatment with 0.5 µM antimycin A. PCP import is affected by antimycin A when ATP is depleted but not under standard conditions. Luminescence was read from 125 µl reactions in a white round-bottom 96 well plate (Thermo Scientific) on either a CLARIOStar Plus (BMG LABTECH), or a BioTek Synergy Neo2 plate reader (BioTek Instruments) without emission filters. Measurements were taken every 6 s or less, and acquisition time was either 0.1 s (on the CLARIOStar Plus reader) or 0.2 s (on the Synergy Neo2 reader).

## Western blot-based import assay

Mitochondria were resuspended to 0.5 mg/ml in import buffer lacking Prionex reagent and supplemented with 2 mM NADH, 1 mM ATP, 0.1 mg/ml creatine kinase, and 5 mM phosphocreatine. Import substrates were prepared with DTT, and with or without MTX and NADPH accordingly, as described above, then added to import reactions for a concentration of 40 nM. Reactions were for 15 min at 25°C, and were stopped by the addition of a mixture of valinomycin (1 μM), oligomycin (4 μM), and antimycin A (8 μM) and transfered to ice. These samples were divided in half and one portion of each treated with proteinase K (40 μg/ml) for 20 min at 4°C. The protease treatment was stopped with PMSF (3 mM). Mitochondria were collected by centrifugation (16,000× $g$, 15 min, 4°C), washed with SM buffer containing 2 mM PMSF, and collected by centrifugation as before.

## Estimation of mitochondrial matrix volume

The mitochondrial matrix volume as a fraction of reaction volume was estimated using the previously published yeast mitochondrial matrix volume of 1.62±0.3 μl/mg (*Koshkin and Greenberg, 2002*). Thus when mitochondria are at 50 μg/ml, matrix volume will be 81±15 nl/ml or ~1/12345.68 total volume (between 1/15151.5 and 1/10416.7 accounting for error).

## Data processing and analysis

NanoLuc assay data were processed using a combination of software: Microsoft Excel, pro Fit 7, and GraphPad Prism versions 8 and 9. Data were then normalised to the maximum luminescence measurement for each experiment.

In most cases, the resulting data were fitted using pro Fit to a model for two consecutive, irreversible steps, where the final one gives rise to a signal (*Fersht, 1984*):

$$Y = A_0 \left( 1 + \tfrac{1}{k_1 - k_2} \left( k_2 e^{-k_1 t} - k_1 e^{-k_2 t} \right) \right)$$

where $A_0$ is the amplitude, $k_1$ and $k_2$ are the two rate constants, Y is the signal, and t is the time. Note that this equation produces the same result whichever order $k_1$ and $k_2$ are in. Subsequent analyses of the resultant data were done in GraphPad Prism; linear and non-linear (Michaelis-Menten) regression. Values for $k_1$ were capped at 30, as times faster could not reasonably be resolved.

## Membrane potential measurements with isolated mitochondria

Isolated mitochondria were diluted to 50 μg/ml in import buffer (described above) supplemented with 1 mM ATP, 0.1 mg/ml creatine kinase, 5 mM phosphocreatine, 10 μM GST-Dark protein and 0.5 μM TMRM. Relative $\Delta\psi$ was monitored over time as a change in fluorescence of the $\Delta\psi$-dependent dye TMRM in quenching mode. Fluorescence was measured at an excitation wavelength of 548 nm and an emission wavelength of 574 nm, in black plates, on a BioTek Synergy Neo2 plate reader (BioTek Instruments). The inner membrane PMF was generated by injecting 2 mM NADH, and PCP proteins added manually after stabilisation of fluorescence. Depolarisation was confirmed at the end of the assay by injecting the protonophore CCCP.

# Acknowledgements

We would like to thank Prof. Andrew Halestrap for his insight and enthusiastic discussions on the mysteries of mitochondrial bioenergetics. We also thank past and present members of the Collinson lab who helped to get this project off the ground, particularly Drs Andrew Richardson and Dylan Noone. Funding: This research was funded by the Wellcome Trust: Investigator Award to IC (104632/Z/14/Z).

# Additional information

## Funding

| Funder | Grant reference number | Author |
|---|---|---|
| Wellcome Trust | 104632 | Holly C Ford<br>William J Allen<br>Gonçalo C Pereira<br>Xia Liu<br>Ian Collinson |

The funders had no role in study design, data collection and interpretation, or the decision to submit the work for publication. For the purpose of Open Access, the authors have applied a CC BY public copyright license to any Author Accepted Manuscript version arising from this submission.

## Author contributions

Holly C Ford, William J Allen, Conceptualization, Data curation, Formal analysis, Investigation, Methodology, Writing - original draft, Writing – review and editing; Gonçalo C Pereira, Conceptualization, Methodology, Writing – review and editing; Xia Liu, Methodology; Mark Simon Dillingham, Validation, Writing – review and editing; Ian Collinson, Conceptualization, Investigation, Project administration, Resources, Supervision, Writing - original draft, Writing – review and editing

## Author ORCIDs

Holly C Ford ![ORCID] http://orcid.org/0000-0003-1400-8049
William J Allen ![ORCID] http://orcid.org/0000-0002-9513-4786
Gonçalo C Pereira ![ORCID] http://orcid.org/0000-0001-9638-0615
Mark Simon Dillingham ![ORCID] http://orcid.org/0000-0002-4612-7141
Ian Collinson ![ORCID] http://orcid.org/0000-0002-3931-0503

## Decision letter and Author response

Decision letter https://doi.org/10.7554/eLife.75426.sa1
Author response https://doi.org/10.7554/eLife.75426.sa2

# Additional files

## Supplementary files

• Supplementary file 1. Table containing full list of PCPs, their amino acid sequences, and expression vectors.

• Transparent reporting form

## Data availability

We present secondary analysis of raw optical readout data. All raw data is included in the manuscript, supplementary information and source data.

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
