## [Editor Report]

In this study, a bioluminescence-based technique is used to analyze the import of precursor proteins into the mitochondrial matrix in real time. This is an innovative technical advance that can provide mechanistic details of the kinetic steps involved in mitochondrial protein import. It may potentially be used for other membrane protein transport systems and for drug screening studies targeting the mitochondrial import apparatus.

---

## [Decision Letter]

**Decision letter after peer review:**

Thank you for submitting your article "Towards a molecular mechanism underlying mitochondrial protein import through the TOM and TIM23 complexes" for consideration by *eLife*. Your article has been reviewed by 3 peer reviewers, and the evaluation has been overseen by Klaus Pfanner as the Reviewing Editor and Vivek Malhotra as the Senior Editor. The following individuals involved in review of your submission have agreed to reveal their identity: Raffaele Ieva (Reviewer #2); Nathan N Alder (Reviewer #3).

Essential revisions:

1) The following points should be addressed by additional experimental data and text modification to substantiate the findings (detailed in the reports of the reviewers):

a) Provide additional experimental evidence for the attribution of the two rate limiting steps: for instance, elucidation of the k1 step using TOM mutants and/or protease pre-treatment, or elucidation of the k2 step using mitoplasts.

b) Demonstrate that the DHFR-containing PCP used in this study are imported in the absence of MTX (standard approaches to measure the amount of PCP that becomes inaccessible to external proteases) and that translocation intermediates of DHFR-containing substrates are indeed stabilized in the presence of MTX.

In the case of a very efficient block for PCP-pep86-DHFR, the speculation about the reaction being "single turnover" in vitro would make good sense. If not, this speculation should be removed.

c) Use protonophore titration or direct readouts of the membrane potential (e.g., by TMRM) to clarify and support the conclusions of Figures 4 and 5. It should be clarified if the import system disrupts the membrane potential.

2) The following points should be clarified or modified in the text (detailed in the reports of the reviewers).

a) Clarification about the single-turnover nature of this experimental system, and perhaps that other factors could contribute to what look like single-turnover conditions.

b) Further discussion of the membrane potential collapse that appears to occur with imported substrates.

c) Further discussion of the implications and assumptions of two sequential transport processes (TOM, then TIM23) vs. coupled TOM/TIM23 transport.

d) Clarification on data fitting for luminescence time courses that do not conform to a simple model with two rate constants.

e) Discussion on the potential confounding effects of longer PCPs, and why they need higher concentration to reach maximal amplitude.

f) Clarification of the Discussion section (page 15, lines 23-27) describing the stoichiometric limitation of protein import by the number of TIM23 complexes.

*Reviewer #1 (Recommendations for the authors):*

Ford et al., investigated protein import into the mitochondrial matrix via the presequence pathway using an innovative NanoLuc translocation assay. In this assay, a model precursor tagged C-terminally with a small fragment of the NanoLuc enzyme is imported into purified mitochondria. The mitochondria were prepared from a yeast strain overexpressing a NanoLuc enzyme lacking the small fragment that contains a mitochondrial presequence that directs the protein into the matrix. Upon import of the model precursor, the active NanoLuc enzyme is formed and produces a uminescence signal in the present of a dye. The authors used this assay to study the effect of ATP and loss of the membrane potential on the kinetics of protein import. The kinetic profiles indicate the presence of two rate-limiting steps. The authors propose that the first step corresponds to binding of the precursor protein to the TOM complex. The second step could represent the initiation of transport across the inner membrane. They further found that precursor properties such as net charge and size have an impact on these steps. Based on the findings the authors proposed a kinetic model including two rate-limiting steps.

The presented assay is elegant and the findings are potentially interesting. Certain control experiments are required to substantiate some of the reported findings.

Specific points:

The authors should provide additional evidence that k1 corresponds to precursor binding to the TOM complex. For instance, is k1 affected when mitochondria from TOM mutants were used or mitochondria were treated with proteinase K before the import reaction?

The authors should show via Western blotting that the precursor proteins are imported into mitochondria, e.g. showing that processing of the precursor protein occurs. The authors should also provide evidence that the DHFR-fused precursor proteins are stalled during translocation (Figure 2).

In Figure 5D the authors found that the NanoLuc import assay dissipates the membrane potential, even when the wild-type protein was used? Is this a general problem with this assay? The authors should discuss advantages and disadvantages to the well-established import assay with radiolabelled precursor proteins into isolated mitochondria.

The authors should improve the labelling of the figures. For instance, the different colors in plots should be defined in the figure. What is the meaning of the number on top of the gel in Figure S1A?

*Reviewer #2 (Recommendations for the authors):*

I suggest to clarify the following points:

– The results obtained with PCP-pep86-DHFR lead the authors to conclude that “The import reaction is largely single turnover under the experimental conditions deployed”. This statement is based on the fact that saturating amounts of PCP-pep86-DHFR are imported with similar efficiency irrespective of whether MTX (that should induce a block of DHFR transport) is supplemented to the import reaction. It remains however unclear whether, under these experimental conditions, folding of DHFR is efficiently stabilized by MTX, blocking transport. This is not the case with the similar construct PCP-DHFR-pep86, given that approximately one third of the protein can be imported in the presence of MTX. Whether PCP-pep86-DHFR is folded and blocked at import sites remains to be determined.

– The observation that positively charged preproteins are imported with a kinetics characterized by a single rate-limiting step seems compatible with a mechanism of transport via the coupled activity of the TOM and TIM23 translocases and formation of a two-membrane spanning protein transport intermediate. The authors should consider the possibility that these positively charged PCPs might be transported via the TOM-TIM23 supercomplex, as previously shown for Cyb2-DHFR fusion constructs that lack the stop-transfer signal.

– The length variants series presents an important variable. By preserving unchanged the PCP MTS and increasing the size of the mature portion of the constructs, the ratio of MTS concentration versus the concentration of mature polypeptide segments decreases. The authors should comment on the possibility that the increased concentration of non-import-competent polypeptide segments (mature regions of the PCPs) may interfere or compete with presequence recognition, providing an alternative explanation as to why longer PCPs require a higher concentration to reach maximal amplitude.

*Reviewer #3 (Recommendations for the authors):*

This is a very powerful and potentially useful approach for the measurement of mitochondrial protein import in real time. In addition to providing additional clarification and explanation for specific points in the Public Review, I would offer the following specific experiments or modifications that could strengthen this study:

1) Make sure to clearly label the MW markers of Figure 1 —figure supplement 1, panel A, along with an explanation of the disparity in sizes between the mitochondrial 11S and the purified 11S standards.

2) Provide direct measurements of membrane potential relevant to Major Comment #8.

3) Conduct control experiments using more traditional approaches for measuring import (Westerns or autoradiography) as described in Major Point #12, as a way of comparing this luminescence-based approach to historically used techniques.

4) Consider conducting imports with mitoplasts as a way to test the kinetic model that includes the TIM23-mediated step alone, as described in Major Point #13.

---

## [Author Response]

Essential revisions:1) The following points should be addressed by additional experimental data and text modification to substantiate the findings (detailed in the reports of the reviewers):a) Provide additional experimental evidence for the attribution of the two rate limiting steps: for instance, elucidation of the k1 step using TOM mutants and/or protease pre-treatment, or elucidation of the k2 step using mitoplasts.

We followed the reviewers’ suggestion to perform additional experiments to confirm the assignments of rate constants *k1* and *k*2. The best approach turned out to be the complete removal of the outer-membrane, rather than more subtle approaches: proteolysis treatment of the outer-membrane was less effective, and analysis of mutants of *tom40* would have taken too long. Therefore, import assays were performed on mitoplasts, on the basis that this should eliminate or change *k*1 while leaving *k*2 unaffected.

We found mild digitonin treatment (2 mg/ ml) to be the most effective way to remove the outer membrane while leaving the mitoplasts intact, as judged by Western blotting (new Figure 1 —figure supplement 1D) and respiration measurements ‒ see new figure (Figure 5 —figure supplement 1A-B).

Import measurements were then performed on these mitoplasts with the shortest and longest PCPs (L and DDDL) over a range of concentrations. Just as expected from the model, *k*2ʹ is virtually identical between intact mitochondria and mitoplasts for both PCPs (see new Figure 5A-B, filled *versus* open brown circles). And although a second rate constant (*k*1') is still required to fit the data, it is much faster in the absence of the OMM (see new Figure 5A-B, open vs filled teal circles). Most likely this new rate constant represents direct binding of the PCP to the TIM23 complex at the outer surface of the IMM.

This new result therefore directly confirms the original assignment of *k*1 taking place at the OMM and *k*2 at the IMM. Furthermore, it supports the conclusion that direct handover from TOM to TIM23 is not absolutely required for import.

b) Demonstrate that the DHFR-containing PCP used in this study are imported in the absence of MTX (standard approaches to measure the amount of PCP that becomes inaccessible to external proteases) and that translocation intermediates of DHFR-containing substrates are indeed stabilized in the presence of MTX.In the case of a very efficient block for PCP-pep86-DHFR, the speculation about the reaction being "single turnover" in vitro would make good sense. If not, this speculation should be removed.

We have performed classical (Western blot) import assays with PCP-pep86-DHFR, in the presence or absence of MTX, both with and without external protease k treatment – see new figure (Figure 2 —figure supplement 1).

In the absence of MTX, the PCP is imported efficiently as expected: the presequence is processed and the mature protein protected from protease. In the presence of MTX, the PCP imports far enough for presequence processing but remains completely susceptible to protease K – (see middle panel, red box). This confirms that DHFR + MTX is an efficient block of pre-protein transport.

Furthermore, although the calculations are approximate, the amount of imported PCPpep86-DHFR in the absence of MTX corresponds closely with the expected number of Tim23 import sites (right panel) – lending further support to the idea that transport is single turnover.

c) Use protonophore titration or direct readouts of the membrane potential (e.g., by TMRM) to clarify and support the conclusions of Figures 4 and 5. It should be clarified if the import system disrupts the membrane potential.

As requested, we have added valinomycin titrations to show that 10 nM effectively depletes membrane potential (see Figure 4 —figure supplement 1A) and reduces import of all pre-proteins without completely abolishing it (Figure 4 —figure supplement 1B). As has now been clarified, this was our initial rationale for choosing 10 nM valinomycin.

Our conclusion is that protein import itself consumes Δψ in a manner that is dependent on PCP net charge (see what is now Figure 6D). As there is no Nano-Glo luciferase assay substrate present in the TMRM assays, it cannot be the NanoLuc measurement system causing this effect. To confirm this further, we have added new data showing that Δψ is depleted even when pep86 is absent from the PCP (Figure 6 —figure supplement 2). Therefore, this observation is not an artefact of the assay system, but a genuine effect of protein import into isolated mitochondria.

2) The following points should be clarified or modified in the text (detailed in the reports of the reviewers).a) Clarification about the single-turnover nature of this experimental system, and perhaps that other factors could contribute to what look like single-turnover conditions.

Further evidence for this has been provided – see response 1b.

b) Further discussion of the membrane potential collapse that appears to occur with imported substrates.

We have discussed this further and included further data in Figure 4 —figure supplement 1 demonstrating the correlation between Δψ and signal amplitude.

c) Further discussion of the implications and assumptions of two sequential transport processes (TOM, then TIM23) vs. coupled TOM/TIM23 transport.

Further discussion has been added.

d) Clarification on data fitting for luminescence time courses that do not conform to a simple model with two rate constants.

We have rewritten the section where these first occur to make it clearer.

e) Discussion on the potential confounding effects of longer PCPs, and why they need higher concentration to reach maximal amplitude.

We are not entirely sure why this is the case, but we have clarified the discussion on this point.

f) Clarification of the Discussion section (page 15, lines 23-27) describing the stoichiometric limitation of protein import by the number of TIM23 complexes.

This comment has been removed.

Reviewer #1 (Recommendations for the authors):Ford et al., investigated protein import into the mitochondrial matrix via the presequence pathway using an innovative NanoLuc translocation assay. In this assay, a model precursor tagged C-terminally with a small fragment of the NanoLuc enzyme is imported into purified mitochondria. The mitochondria were prepared from a yeast strain overexpressing a NanoLuc enzyme lacking the small fragment that contains a mitochondrial presequence that directs the protein into the matrix. Upon import of the model precursor, the active NanoLuc enzyme is formed and produces a luminescense signal in the present of a dye. The authors used this assay to study the effect of ATP and loss of the membrane potential on the kinetics of protein import. The kinetic profiles indicate the presence of two rate-limiting steps. The authors propose that the first step corresponds to binding of the precursor protein to the TOM complex. The second step could represent the initiation of transport across the inner membrane. They further found that precursor properties such as net charge and size have an impact on these steps. Based on the findings the authors proposed a kinetic model including two rate-limiting steps.The presented assay is elegant and the findings are potentially interesting. Certain control experiments are required to substantiate some of the reported findings.Specific points:The authors should provide additional evidence that k1 corresponds to precursor binding to the TOM complex. For instance, is k1 affected when mitochondria from TOM mutants were used or mitochondria were treated with proteinase K before the import reaction?

We now include additional evidence that *k*1ʹ corresponds to a step preceding IMM transport. When PCPs are imported into mitoplasts instead of mitochondria, *k*1ʹ but not *k*2ʹ is affected.

The authors should show via Western blotting that the precursor proteins are imported into mitochondria, e.g. showing that processing of the precursor protein occurs. The authors should also provide evidence that the DHFR-fused precursor proteins are stalled during translocation (Figure 2).

This is now included (Figure 2 —figure supplement 1).

In Figure 5D the authors found that the NanoLuc import assay dissipates the membrane potential, even when the wild-type protein was used? Is this a general problem with this assay? The authors should discuss advantages and disadvantages to the well-established import assay with radiolabelled precursor proteins into isolated mitochondria.

It is not the NanoLuc import assay that dissipates the membrane potential, but rather protein import itself. See response 1c.

The authors should improve the labelling of the figures. For instance, the different colors in plots should be defined in the figure. What is the meaning of the number on top of the gel in Figure S1A?

We have amended the figures as requested.

Reviewer #2 (Recommendations for the authors):I suggest to clarify the following points:– The results obtained with PCP-pep86-DHFR lead the authors to conclude that "The import reaction is largely single turnover under the experimental conditions deployed". This statement is based on the fact that saturating amounts of PCP-pep86-DHFR are imported with similar efficiency irrespective of whether MTX (that should induce a block of DHFR transport) is supplemented to the import reaction. It remains however unclear whether, under these experimental conditions, folding of DHFR is efficiently stabilized by MTX, blocking transport. This is not the case with the similar construct PCP-DHFR-pep86, given that approximately one third of the protein can be imported in the presence of MTX. Whether PCP-pep86-DHFR is folded and blocked at import sites remains to be determined.

Using a classic Western blot-based import assay, we have provided evidence that folding of DHFR is stabilised by MTX (Figure 2 —figure supplement 1).

– The observation that positively charged preproteins are imported with a kinetics characterized by a single rate-limiting step seems compatible with a mechanism of transport via the coupled activity of the TOM and TIM23 translocases and formation of a two-membrane spanning protein transport intermediate. The authors should consider the possibility that these positively charged PCPs might be transported via the TOM-TIM23 supercomplex, as previously shown for Cyb2-DHFR fusion constructs that lack the stop-transfer signal.

A discussion of this possibility has now been included.

– The length variants series presents an important variable. By preserving unchanged the PCP MTS and increasing the size of the mature portion of the constructs, the ratio of MTS concentration versus the concentration of mature polypeptide segments decreases. The authors should comment on the possibility that the increased concentration of non-import-competent polypeptide segments (mature regions of the PCPs) may interfere or compete with presequence recognition, providing an alternative explanation as to why longer PCPs require a higher concentration to reach maximal amplitude.

A discussion of this possibility has now been included.

Reviewer #3 (Recommendations for the authors):This is a very powerful and potentially useful approach for the measurement of mitochondrial protein import in real time. In addition to providing additional clarification and explanation for specific points in the Public Review, I would offer the following specific experiments or modifications that could strengthen this study:1) Make sure to clearly label the MW markers of Figure 1 —figure supplement 1, panel A, along with an explanation of the disparity in sizes between the mitochondrial 11S and the purified 11S standards.

As requested, molecular weight markers have now been labelled and an explanation of the 11S size difference included in the text.

2) Provide direct measurements of membrane potential relevant to Major Comment #8.

These measurements were carried out as requested (Figure 4 —figure supplement 1A).

3) Conduct control experiments using more traditional approaches for measuring import (Westerns or autoradiography) as described in Major Point #12, as a way of comparing this luminescence-based approach to historically used techniques.

We performed a Western blot-based import assay with PCP-pep86-DHFR to validate the experiments in Figure 2 (Figure 2 —figure supplement 1).

4) Consider conducting imports with mitoplasts as a way to test the kinetic model that includes the TIM23-mediated step alone, as described in Major Point #13.

We carried out these experiments as requested (Figure 5), and the results support our previous conclusions.